# Senataxin and RNase H2 act redundantly to suppress genome instability during class switch recombination

Hongchang Zhao[1], Stella R Hartono[2], Kirtney Mae Flores de Vera[1], Zheyuan Yu[1,3], Krishni Satchi[1], Tracy Zhao[1], Roger Sciammas[4], Lionel Sanz[2], Frédéric Chédin[2], Jacqueline Barlow[1]*

[1]Department of Microbiology and Molecular Genetics, University of California, Davis, Davis, United States; [2]Department of Molecular and Cellular Biology, University of California, Davis, Davis, United States; [3]Graduate Group in Biostatistics, University of California, Davis, Davis, United States; [4]Center for Immunology and Infectious Diseases, University of California, Davis, Davis, United States

**Abstract** Class switch recombination generates distinct antibody isotypes critical to a robust adaptive immune system, and defects are associated with autoimmune disorders and lymphomagenesis. Transcription is required during class switch recombination to recruit the cytidine deaminase AID—an essential step for the formation of DNA double-strand breaks—and strongly induces the formation of R loops within the immunoglobulin heavy-chain locus. However, the impact of R loops on double-strand break formation and repair during class switch recombination remains unclear. Here, we report that cells lacking two enzymes involved in R loop removal—senataxin and RNase H2—exhibit increased R loop formation and genome instability at the immunoglobulin heavy-chain locus without impacting its transcriptional activity, AID recruitment, or class switch recombination efficiency. Senataxin and RNase H2-deficient cells also exhibit increased insertion mutations at switch junctions, a hallmark of alternative end joining. Importantly, these phenotypes were not observed in cells lacking senataxin or RNase H2B alone. We propose that senataxin acts redundantly with RNase H2 to mediate timely R loop removal, promoting efficient repair while suppressing AID-dependent genome instability and insertional mutagenesis.

*For correspondence: jhbarlow@ucdavis.edu

Competing interest: The authors declare that no competing interests exist.

## Editor's evaluation

R loops have been described at the immunoglobulin heavy chain (Igh) locus long ago. However, their contribution to Igh diversification by class switch recombination (CSR) and locus integrity has been elusive. The authors show that R loop removal by the activity of senataxin and RNase H2 does not influence CSR but is required to suppress genome instability at the Igh locus. This article will be of interest to the audience in the fields of genome integrity and B lymphocyte biology.

## Introduction

Class switch recombination (CSR) is a programmed recombination event in mature B cells that generates antibodies of different isotypes, allowing for their interaction with different effector molecules. Successful CSR is a deletional rearrangement catalyzed by the formation of DNA double-strand breaks (DSBs) within the immunoglobulin heavy-chain (IgH) switch regions. DSBs are formed by a series of events, initiated by the deamination of cytosine residues in single-stranded DNA by activation-induced cytidine deaminase (AID) (*Chaudhuri et al., 2003*; *Muramatsu et al., 2000*; *Pham et al., 2003*;

**eLife digest** The immune system is a complex network of cells and molecules, which helps to protect the body from invaders. The adaptive immune system can recognise millions of assailants, kill them, and 'learn' from this experience to mount an even quicker defence the next time the body is infected.

To achieve this level of protection, specific immune cells, called B cells, divide when they come into contact with a molecule from a foreign particle, the antigen. The cloned B cells then produce millions of protective proteins, the antibodies, which patrol the blood stream and tag harmful particles for destruction.

An antibody resembles a Y-shaped structure that contains a 'variable' region, which gives it the specificity to interact with an antigen, and a 'constant' region, which interacts with components of the immune system and determines the mechanisms used to destroy a pathogen. Based on the constant region, antibodies can be divided into five main classes.

B cells are able to switch their production from one antibody class to another in an event known as class switch recombination, by making changes to the constant region. They do this by cutting out a portion of the genes for the constant region from their DNA and fusing the remaining DNA. The resulting antibodies still recognise the same target, but interact with different components of the immune system, ensuring that all the body's forces are mobilised.

R-loops are temporary structures that form when a cell 'reads' the instructions in its DNA to make proteins. R-loops provide physical support by anchoring the transcription template to the DNA. They help control the activity of genes, but if they stay on the DNA for too long they could interfere with any form of. DNA repair – including the cutting and fusing mechanisms during class switch recombination.

To find out more about this process, Zhao et al. used B-cells from mice lacking two specific proteins that usually help to remove R-loops. Without these proteins, the B cells generated more R-loops than normal. Nevertheless, the B-cells were able to undergo class switch recombination, even though their chromosomes showed large areas of DNA damage, and DNA sections that had been repaired contained several mistakes.

Errors that occur during class switch recombination have been linked to immune disorders and B cell cancers. The study of Zhao et al. shows that even if R-loops do not affect some processes in B cells, they could still impact the overall health of their DNA. A next step would be to test if an inability to remove R-loops could indeed play a role in immune disorders and B-cell cancers.

*Revy et al., 2000*). The resulting U:G mismatches are processed by multiple DNA repair pathways producing mutations or single-strand DNA (ssDNA) breaks (*Stavnezer and Schrader, 2014*). In CSR, multiple single-strand breaks on both Watson and Crick DNA strands create double-stranded breaks (DSBs), which initiate recombination between two adjacent but distinct genomic loci. In successful CSR, non-homologous end-joining (NHEJ) proteins repair the DSBs by joining the two distal DNA ends, deleting the intervening DNA.

## AID targeting during CSR

AID recruitment to chromatin is a highly regulated act as off-target AID activity promotes IgH and non-IgH DSBs and translocations associated with carcinogenesis (*Ramiro et al., 2004*; *Robbiani et al., 2008*; *Robbiani et al., 2009*). Transcriptional activity is necessary for the formation of ssDNA during CSR and directly promotes AID recruitment to transcribing switch regions (*Chaudhuri et al., 2003*; *Zheng et al., 2015*). AID interacts with RNA polymerase II cofactors, including the transcription factor Spt5, as well as the elongation factor complex PAF1, promoting its recruitment to active switch regions (*Pavri et al., 2010*; *Stanlie et al., 2012*; *Willmann et al., 2012*). AID also associates with the ssDNA binding protein RPA (*Chaudhuri et al., 2004*; *Yamane et al., 2011*). Pre-mRNA splicing also plays a role in AID recruitment as depletion of the splicing regulator PTBP2 impairs CSR efficiency (*Nowak et al., 2011*). More recently, researchers have shown that AID shows a binding preference for nucleic acid sequences forming G4 quadruplex structures, which are highly enriched in switch sequences (*Qiao et al., 2017*).

## R loops form at switch regions during CSR

Transcription at the switch region also induces the formation of R loops in vitro and in vivo—three-stranded nucleotide structures where newly synthesized RNA re-anneals to the DNA template (*Aguilera and García-Muse, 2012*; *Daniels and Lieber, 1995*; *Reaban and Griffin, 1990*; *Yu et al., 2003*). While R loops have long been observed at switch regions, their role in CSR remains confounding. The non-template DNA strand of R loops is single-stranded, creating an ideal substrate for AID (*Pham et al., 2003*; *Ramiro et al., 2003*; *Sohail et al., 2003*). R loop formation in switch regions is sequence-dependent and positively correlates with AID deamination, further suggesting that R loops promote AID recruitment (*Huang et al., 2007*). R loops are more stable in GC-rich sequences (*Ginno et al., 2012*; *Kuznetsov et al., 2018*; *Stolz et al., 2019*), and R loop formation in switch region correlates with G density (*Zhang et al., 2014*). Additionally, G clustering promotes R loop formation in cloned switch regions (*Roy et al., 2008*). Consistent with this, the ATP-dependent RNA helicase DDX1 can promote R loop formation in switch regions, promoting AID recruitment and DSB formation (*Ribeiro de Almeida et al., 2018*). Yet how R loops are resolved to promote DSB repair is less clear. Studies examining the RNA exosome indicate it promotes AID targeting to both strands of the DNA by helping remove R loops from the template strand, thereby exposing the DNA for deamination (*Basu et al., 2011*). However, multiple studies have used ectopic expression of RNase H1 nuclease to reduce switch region R loops, with contradictory results. One study found that ectopic RNase H1 expression in mouse CH12 cells or primary B cells reduces CSR to IgA or IgG1, respectively; however, an earlier study also using CH12 cells observed no effect on CSR to IgA (*Parsa et al., 2012*; *Wiedemann et al., 2016*). Further, transgene-driven expression of murine RNase H1 in primary cells had no effect on CSR to any IgG isotype though somatic hypermutation was increased (*Maul et al., 2017*). Thus, the enzymes involved in switch region R loop removal and their impact on CSR remain elusive.

Here, we use mouse knockout models of two key enzymes implicated in R loop removal, the helicase senataxin (SETX) and a subunit of the heterotrimeric nuclease RNase H2, to assess the impact of defective R loop removal on DNA DSB repair during CSR in primary B lymphocytes (*Becherel et al., 2013*; *Hiller et al., 2012*). Both SETX and RNase H2 have been implicated in the resolution of R loops and suppression of genome instability from yeast to humans (*Cristini et al., 2022*; *Mischo et al., 2011*; *Skourti-Stathaki et al., 2011*; *Zhao et al., 2018*). Further, Sen1 and RNase H activity act redundantly to suppress R loops and maintain cell viability in budding yeast (*Costantino and Koshland, 2018*). We found that loss of both SETX and RNase H2 activity (*Setx*[-/-]*Rnaseh2b*[f/f]) exhibits increased R loops specifically at the Sμ switch region in resting or activated B cells; however, loss of SETX or RNase H2B alone did not consistently increase R loops. This increase in R loops correlates with enhanced genome instability at the heavy-chain locus as ~10% of double-deficient B cells contain persistent IgH breaks and translocations. We also observed increased mutations and insertion events in SETX- and RNase H2-deficient B cells by molecular analysis of switch junctions, though there was no defect in CSR efficiency. Taken together, our data suggest that timely R loop removal at switch regions by a SETX/RNase H2 mechanism during CSR suppresses error-prone end-joining and translocation formation at IgH.

## Results

### RNA:DNA hybrids are increased in *Setx*[-/-]*Rnaseh2b*[f/f] cells

To determine the effect of R loop metabolism on CSR, we generated mice lacking two enzymes involved in R loop removal, SETX and RNase H2. RNase H2 activity is essential for mouse embryogenesis; therefore, mice harboring a conditional *Rnaseh2b* allele were crossed to *Cd19*[Cre] for B-cell-specific gene deletion (*Rnaseh2b*[f/f]), then crossed with mice containing a germline deletion of *Setx* (*Setx*[-/-]) (*Figure 1A*; *Becherel et al., 2013*; *Hiller et al., 2012*). Cre-mediated deletion of *Rnaseh2b* resulted in 80–90% deletion of the genomic DNA and a fivefold reduction in transcription (*Figure 1—figure supplement 1A and B*). Freshly isolated splenic B cells were stimulated with LPS, IL-4, and α-RP105 to induce CSR to IgG1. Then, 72 hr post-stimulation, genomic DNA was isolated and R loop levels were measured by dot blot probed with the RNA:DNA hybrid-specific antibody S9.6 (*Figure 1B*). Cells lacking both SETX and RNase H2B showed a fourfold increase in total R loops, while loss of SETX or RNase H2B alone showed no significant change in R loop levels (*Figure 1B and C*). To determine whether R loops were increased within the switch regions,

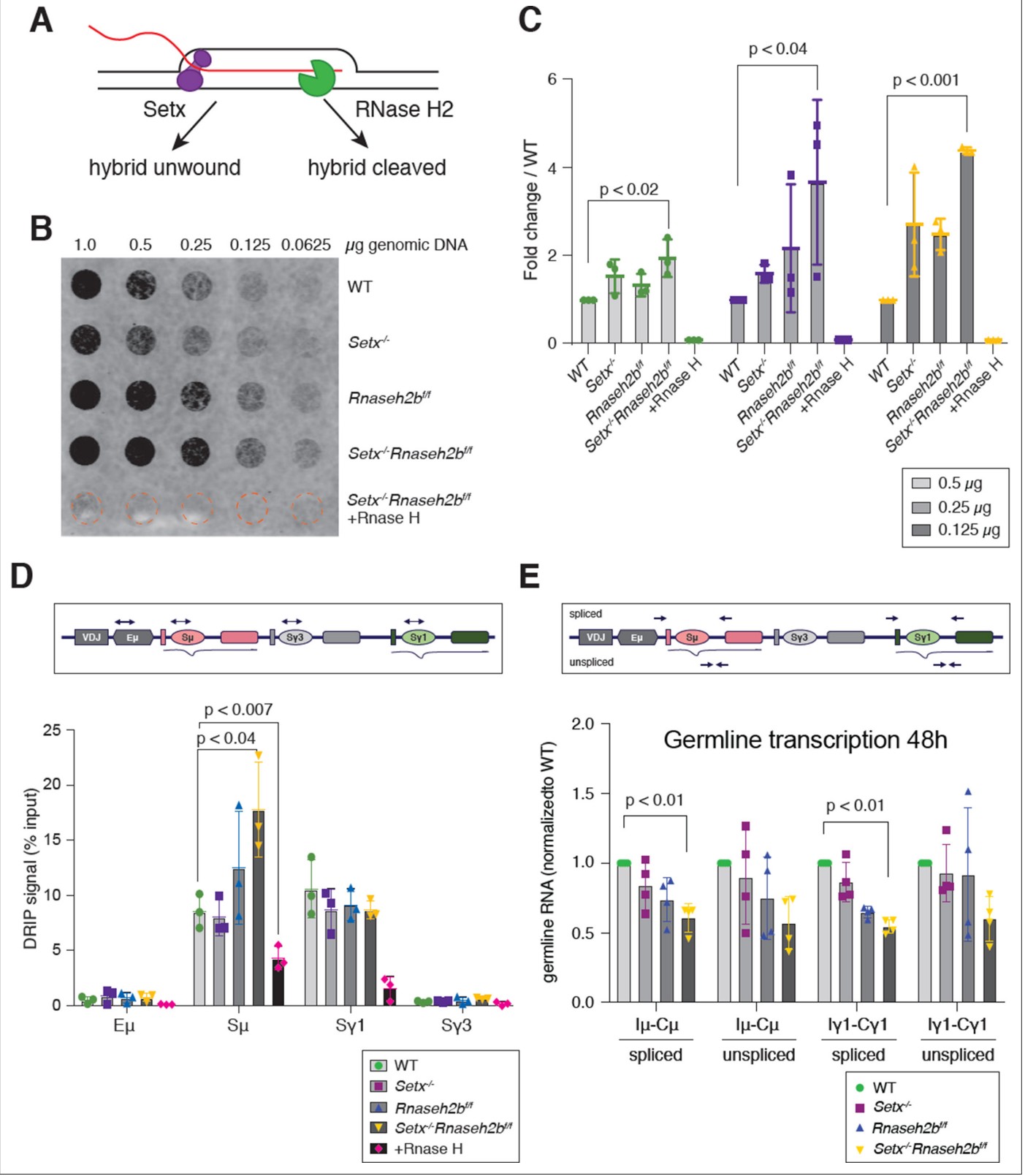

**Figure 1.** RNA-DNA hybrids are increased in *Setx⁻/⁻Rnaseh2b^f/f* B cells during class switch recombination (CSR) to IgG1. (**A**) Senataxin (SETX) and RNase H2 contribute to RNA:DNA hybrid removal; SETX helicase activity unwinds the nucleotide strands retaining both the RNA and DNA components while RNase H2 cleaves RNA, retaining only the DNA strand. (**B**) Dot blot analysis of R loop formation: twofold serial dilutions of genomic DNA starting at 1 µg were arrayed on a nitrocellulose membrane and probed using the S9.6 antibody; RNase H treatment of the *Setx⁻/⁻Rnaseh2b^f/f* sample was used as the

*Figure 1 continued on next page*

*Figure 1 continued*

negative control. (**C**) Quantitation of 0.5 µg, 0.25 µg, and 0.125 µg dot blot from (**B**) using ImageJ; values were normalized to WT, set as 1. Figures are expressed as fold change relative to WT. Error bars are the standard deviation between different experiments; *p<0.05 comparing different genotypes using multiple *t*-test (n = 3 mice/genotype). (**D**) Diagram of the IgH locus showing the location of PCR products used for chromatin immunoprecipitation (ChIP) and DNA:RNA hybrid immunoprecipitation (DRIP) assays. DRIP assay was performed with S9.6 antibody using primary B cells after 72 hr of stimulation to IgG1 with LPS/IL-4/α-RP105; RNase H treatment of *Setx⁻/⁻Rnaseh2b^{f/f}* sample was used as the negative control. Relative enrichment was calculated as ChIP/input, and the results were replicated in three independent experiments. Error bars show standard deviation; statistical analysis was performed using one-way ANOVA (n = 3 mice/genotype). (**E**) Diagram of the IgH locus showing the location of PCR products for germline transcription under IgG1 stimulation with LPS/IL-4/α-RP105. Real-time RT-PCR analysis for germline transcripts (Ix-Cx) at donor and acceptor switch regions in WT, *Setx⁻/⁻*, *Rnaseh2b^{f/f}*, and *Setx⁻/⁻ Rnaseh2b^{f/f}* splenic B lymphocytes cultured for 48 hr with LPS/IL-4/α-RP105 stimulation. Expression is normalized to CD79b and is presented as relative to expression in WT cells, set as 1. Error bars show standard deviation; statistical analysis was performed using multiple *t*-test (n = 4 mice/genotype).

The online version of this article includes the following source data and figure supplement(s) for figure 1:

**Source data 1.** Uncropped raw dot blot analysis of R loop formation.

**Source data 2.** Numerical data used to generate graphs in *Figure 1C–E*.

**Figure supplement 1.** *Rnaseh2b* deletion efficiency by CD19-cre.

**Figure supplement 1—source data 1.** Uncropped image of PCR analysis of Cre-mediated *Rnaseh2b* deletion.

**Figure supplement 1—source data 2.** Numerical data used to generate graphs in *Figure 1—figure supplement 1A–D*.

**Figure supplement 2.** DNA:RNA hybrid formation at the *Actb* locus.

**Figure supplement 2—source data 1.** Numerical data used to generate graph in *Figure 1—figure supplement 2B*.

we used DNA:RNA hybrid immunoprecipitation (DRIP) using a monoclonal antibody specific for DNA:RNA hybrids, S9.6. We found a consistent increase in R loop abundance at Sµ specifically in *Setx⁻/⁻Rnaseh2b^{f/f}* cells, while single mutants exhibited R loop levels similar to WT cells at 72 hr post-stimulation during active DNA repair (*Figure 1D*). We also observed DRIP signal at the Sγ1 switch region; however, all genotypes exhibited similar levels (*Figure 1D*). As Sµ is highly transcribed in naïve B cells, we also measured DRIP signal in resting B cells prior to stimulation (*Figure 1—figure supplement 1C*) and again found elevated DRIP signal specifically in in *Setx⁻/⁻Rnaseh2b^{f/f}* cells. We observed almost no DRIP signal in any genotype at Sγ1, consistent with prior observations that R loop levels correlate with transcriptional activation (*Ginno et al., 2012*; *Sanz et al., 2016*; *Yu et al., 2003*). From these results, we conclude that SETX and RNase H2 act redundantly to remove RNA:DNA hybrids at Sµ.

To assess the impact of SETX and RNase H2B loss on an independent locus, we next examined RNA:DNA hybrid signal along the beta-actin gene locus *Actb*. By DRIP-seq, we found the RNA:DNA hybrid footprint to be largely similar between the four genotypes (*Figure 1—figure supplement 2A*). DRIP-qPCR analysis showed a significant increase in signal at the *Actb* intron 2 locus only in *Setx⁻/⁻Rnaseh2b^{f/f}* cells; however, all other loci examined showed similar RNA:DNA hybrid levels in all genotypes examined (*Figure 1—figure supplement 2B*). Of note, we also observed a trend for reduced DRIP signal in *Setx⁻/⁻* cells at the promoter and terminator regions. While somewhat surprising, these results are consistent with published reports showing depletion of SETX decreased R loops genome-wide including at the *ACTB* locus (*Richard et al., 2020*). Together, these results suggest that SETX and RNase H2 impact R loop levels at discrete loci.

## Transcription is not increased in *Setx⁻/⁻Rnaseh2b^{f/f}* cells

In human cells, the majority of R loop formation positively correlates with gene expression, suggesting that increased transcription can enhance R loop formation (*Sanz et al., 2016*). To determine whether the increased level of R loops at switch regions was due to changes in transcriptional activity, we measured germline transcript levels at switch regions by RT-qPCR 48 and 72 hr post-stimulation. In response to LPS/IL-4/α-RP105, we observed similar levels of unspliced germline Sµ and Sγ1 transcripts in all four genotypes (*Figure 1E*, *Figure 1—figure supplement 1D*). Thus, elevated transcription cannot explain the increased R loop levels at Sµ. Intriguingly, we observed a consistent decrease in spliced Sµ transcripts specifically in *Setx⁻/⁻Rnaseh2b^{f/f}* cells, suggesting that increased or persistent R loop formation may reduce splicing efficiency of germline transcripts specifically at Sµ (*Figure 1E*).

## Cells lacking SETX or RNase H2 are proficient for class switch recombination

R loop formation positively correlates with CSR and is predicted to promote AID recruitment within switch regions, thereby promoting DSB formation (*Huang et al., 2007*; *Yu et al., 2003*). Splicing of germline transcripts also correlates with productive CSR; therefore, it is possible CSR would be reduced in *Setx⁻ᐟ⁻Rnaseh2b^{f/f}* cells (*Hein et al., 1998*; *Marchalot et al., 2020*). To determine whether CSR levels are altered in SETX- and RNaseH2-deficient cells, we measured cell surface expression of IgG1 by flow cytometry. We found that the percent of cells undergoing CSR in response to LPS/IL-4/α-RP105 was similar to WT in *Setx⁻ᐟ⁻*, *Rnaseh2b^{f/f}*, and *Setx⁻ᐟ⁻Rnaseh2b^{f/f}* cells at 72 and 96 hr post-stimulation (*Figure 2A–C*). Thus, neither the observed increase in DNA:RNA hybrid abundance by DRIP nor the reduction in splicing at Sμ significantly impacts CSR to IgG1 in *Setx⁻ᐟ⁻Rnaseh2b^{f/f}* cells. Co-stimulation with α-RP105 can speed proliferation and reduce cell death during ex vivo stimulation, potentially obscuring subtle changes in CSR. Cells stimulated with LPS/IL-4 alone also showed no difference in CSR between the four genotypes, indicating indeed there is no major defect in CSR efficiency from loss of Setx and RNase H2 activity (*Figure 2—figure supplement 1*). To determine whether CSR is proficient to other isotypes, we next measured CSR to IgA and IgG2b by stimulating cells with either LPS/α-RP105/TGF-β/CD40L or LPS/α-RP105/TGF-β, respectively. We found that switching to IgA and IgG2b was also at WT levels in *Setx⁻ᐟ⁻*, *Rnaseh2b^{f/f}*, and *Setx⁻ᐟ⁻Rnaseh2b^{f/f}* cells (*Figure 2D–G*). Together, these results indicate that loss of SETX and RNaseH2 does not impact overall CSR efficiency.

## *Setx⁻ᐟ⁻Rnaseh2b^{f/f}* cells have persistent unrepaired DNA damage at IgH

Defects in DNA repair during CSR result in genome instability at IgH visible as persistent DSBs and translocations in miotic chromosome spreads. Defects in R loop removal also correlate with increased DNA damage; therefore, we tested whether loss of SETX and RNase H2 increased genome instability at IgH. To measure persistent DNA damage and translocations at IgH, we performed fluorescent in situ hybridization (FISH) 72 hr post-stimulation in cells switching to IgG1. Spontaneous damage observed in *Setx⁻ᐟ⁻* and *Rnaseh2b^{f/f}* cells was similar to WT levels; however, *Setx⁻ᐟ⁻Rnaseh2b^{f/f}* cells harbored significant DNA damage, including chromosome fusions (*Figure 3A*). These results are similar to reports in budding yeast where combined loss of Sen1 and RNase H activity resulted in a synergistic increase in DNA damage (*Costantino and Koshland, 2018*). No IgH breaks were observed in WT cells and were only found occasionally in *Setx⁻ᐟ⁻* and *Rnaseh2b^{f/f}* cells (*Figure 3B*). However, we consistently observed IgH breaks or translocations in ~10% of *Setx⁻ᐟ⁻Rnaseh2b^{f/f}* cells (*Figure 3B and C*, n = 4 mice).

## B cell development in the bone marrow is normal in cells lacking SETX or RNase H2

Single-cell RNA sequencing studies have detected CD19 expression as early as the pre-pro stage in B cell development, and CD19-cre-driven deletion events can be visualized in pro- and pre-B cell stages by the ROSA26-EYFP marker (*Morgan and Tergaonkar, 2022*; *Yasuda et al., 2021*). To determine whether loss of Setx, RNase H2, or both proteins altered lymphocyte development, we analyzed B cell progenitors in the bone marrow (BM). BM was harvested from WT, *Setx⁻ᐟ⁻*, *Rnaseh2b^{f/f}*, and *Setx⁻ᐟ⁻Rnaseh2b^{f/f}* mice and analyzed by flow cytometry. Quantification of absolute numbers showed that pre-pro B cells, pro-B cells, large pre-B cells, small pre-B cells, and immature B cells were comparable between four genotypes, indicating that B cell development is normal in all genotypes examined (*Figure 3—figure supplement 1A and B*). Quantification of mature naïve B cells isolated from spleen also showed no differences between the four genotypes, supporting the conclusion that depletion of Setx or RNase H2 does not affect B cell development (*Figure 3—figure supplement 1C*). To assess the specificity of Cre deletion, we also plotted CD11B+BM myeloid cells that have no CD19 expression; again cell numbers were comparable between different genotypes (*Figure 3—figure supplement 1D*).

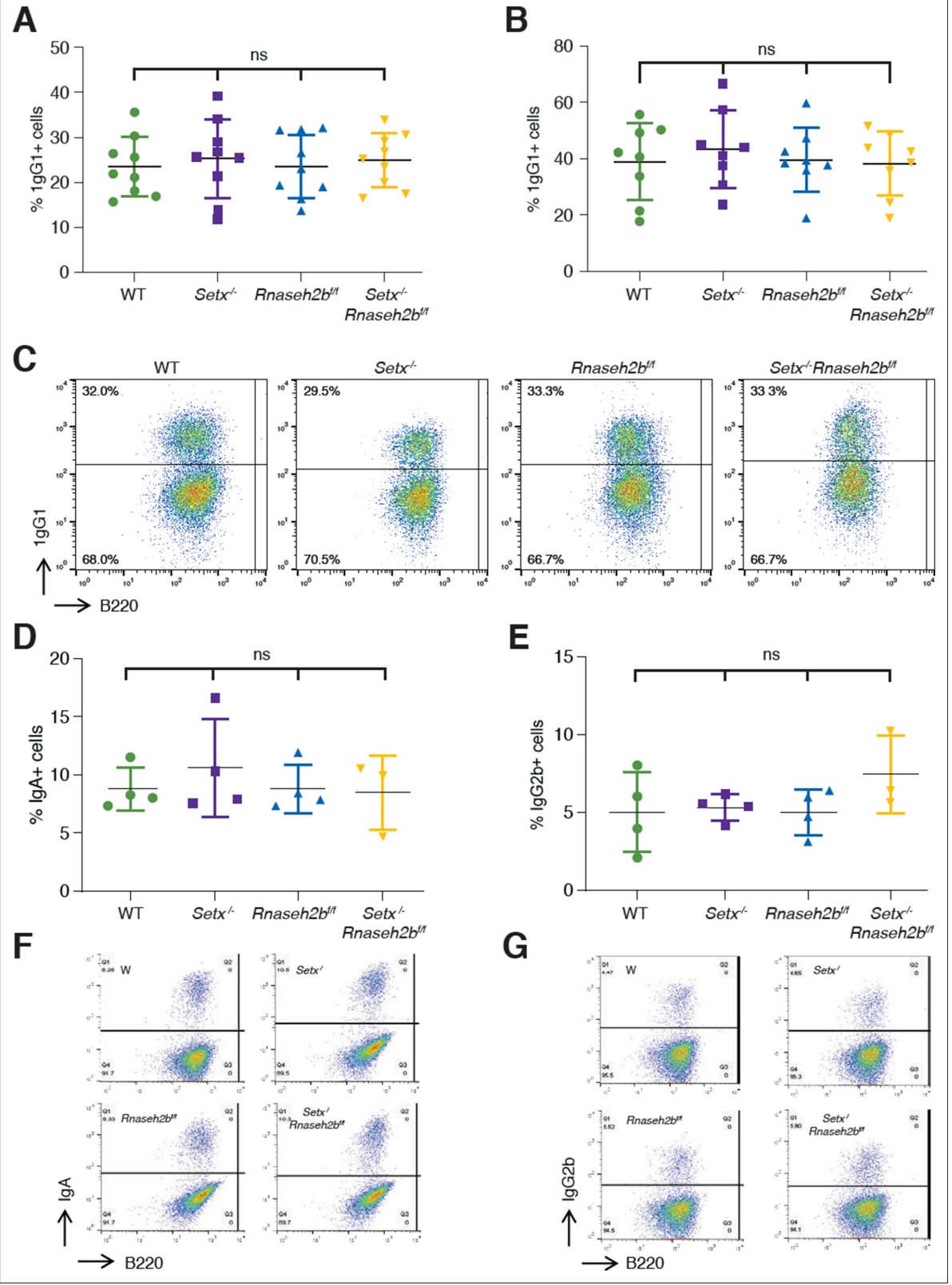

**Figure 2.** Class switch recombination (CSR) is not reduced in *Setx⁻/⁻*, *Rnaseh2b^{f/f}*, or *Setx⁻/⁻Rnaseh2b^{f/f}* B cells. Percentage of cells undergoing CSR to IgG1 72 hr (**A**) and 96 hr (**B**) post-stimulation with LPS/IL-4/α-RP105. (**C**) Representative flow cytometry analyses of IgG1⁺ and B220 expression in response to LPS/IL-4/α-RP105. The percentage of IgG1⁺ B cells is indicated at top left. Percentage of cells undergoing CSR to IgA (**D**) or IgG2B (**E**) 72 hr post-stimulation. (**F**) Representative flow cytometry analyses of IgA⁺ and B220 expression in response to LPS/α-RP105/TGF-B/CD40L. (**G**) Representative

*Figure 2 continued on next page*

*Figure 2 continued*

flow cytometry analyses of IgG2B⁺ and B220 expression in response to LPS/α-RP105/TGF-B. Horizontal lines in dot plots indicate mean, and error bars show standard deviation. Statistical significance versus WT was determined by one-way ANOVA; each dot represents an independent mouse.

The online version of this article includes the following source data and figure supplement(s) for figure 2:

**Source data 1.** Numerical data used to generate graphs in *Figure 2A, B, D and E*.

**Figure supplement 1.** Percentage of cells undergoing class switch recombination (CSR) to IgG1 72 hr post-stimulation with LPS/IL-4 alone.

**Figure supplement 1—source data 1.** Numerical data used to generate graphs in *Figure 2—figure supplement 1*.

## Mis-incorporated ribonucleotides do not contribute to IgH breaks in RNase H2-deficient cells

In addition to cleaving RNA:DNA hybrids, RNase H2 also removes ribonucleotide monophosphates (rNMPs) mis-incorporated into DNA during replication (*Hiller et al., 2012*; *Reijns et al., 2012*; *Williams and Kunkel, 2014*). High levels of genomic rNMPs can lead to genome instability; therefore, they may also contribute to the persistent DNA breaks observed at IgH. We first measured rNMP incorporation by alkaline gel electrophoresis as DNA enriched for rNMPs is sensitive to alkaline hydrolysis, leading to single-strand breaks (*Nick McElhinny et al., 2010*). We found that genomic DNA isolated from $Rnaseh2b^{f/f}$ and $Setx^{-/-}Rnaseh2b^{f/f}$ cells is more sensitive to alkaline treatment than DNA from WT or $Setx^{-/-}$ cells (*Figure 3—figure supplement 2A and B*). Importantly, there was no significant increase in rNMPs in $Setx^{-/-}Rnaseh2b^{f/f}$ cells compared to $Rnaseh2b^{f/f}$ alone. As a control, genomic DNA fragmentation was similar in all four genotypes by native DNA electrophoresis (*Figure 3—figure supplement 2C and D*). These results indicate that rNMPs likely contribute to the total DNA damage observed in $Rnaseh2b^{f/f}$ and $Setx^{-/-}Rnaseh2b^{f/f}$ cells, but rNMPs do not significantly contribute to the IgH breaks observed specifically in $Setx^{-/-}Rnaseh2b^{f/f}$ cells. In the absence of RNase H2, the type I topoisomerase Top1 cleaves rNMPs from genomic DNA (*Huang et al., 2017*; *Williams et al., 2013*). We hypothesized that excess rNMPs would render cells hypersensitive to the Top1 inhibitor camptothecin (CPT), leaving unrepaired breaks in mitosis. Indeed, we found that $Rnaseh2b^{f/f}$ and $Setx^{-/-}Rnaseh2b^{f/f}$ cells were sensitive to CPT treatment, showing similar levels of total genome instability (*Figure 3—figure supplement 2E*). Yet CPT treatment did not increase DNA breaks at IgH in any genotype examined (*Figure 3B* vs. *Figure 3—figure supplement 2F*). We conclude that rNMP mis-incorporation does not substantially contribute to IgH breakage in $Setx^{-/-}Rnaseh2b^{f/f}$ cells even in the presence of exogenous stress.

## Proliferation and cell cycle distribution is normal in cells lacking SETX and RNase H2B

Activated B lymphocytes proliferate extremely rapidly, potentially impacting DNA repair and genome instability (*Lyons and Parish, 1994*). To determine whether cell proliferation is altered, we isolated resting lymphocytes, labeled them with CFSE to track cell division, and stimulated switching to IgG1. After 72 hr, we analyzed CFSE dye dilution and IgG1 expression by flow cytometry. We found that $Rnaseh2b^{f/f}$ and $Setx^{-/-}Rnaseh2b^{f/f}$ cells exhibited a modest decrease in cell proliferation (*Figure 3—figure supplement 3A*). CSR frequency correlates with cell division rate (*Hodgkin et al., 1996*) however, the CSR frequency was similar in all genotypes examined (*Figure 3—figure supplement 3B*). It is possible that the persistent DSBs observed in $Setx^{-/-}Rnaseh2b^{f/f}$ cells trigger DNA damage checkpoint activation, altering cell cycle distribution. Cell cycle phase impacts DSB end resection and repair pathway choice; therefore, we analyzed cell cycle profiles by PI staining (*Symington, 2016*). We found that all genotypes had similar fractions of cells in G1 and S phase cells (*Figure 3—figure supplement 3C and D*). G2/M cells were modestly increased in $Rnaseh2b^{f/f}$ and $Setx^{-/-}Rnaseh2b^{f/f}$ cells, similar to prior reports showing that cells lacking RNase H2 accumulate in G2/M (*Hiller et al., 2012*; *Figure 3—figure supplement 3C and D*; $p < 0.05$, one-way ANOVA). Since $Setx^{-/-}$ cells did not exhibit an increase in G2/M cells, thus, we conclude that the increased genome instability at IgH is not due to changes in DNA repair from altered cell cycle distribution.

## RNase H2 activity rescues DNA:RNA hybrid levels in stimulated B cells

It is possible that the increased DNA:RNA hybrids observed in $Setx^{-/-}Rnaseh2b^{f/f}$ cells arise indirectly due to changes in gene expression or chromatin accessibility earlier in B cell development as

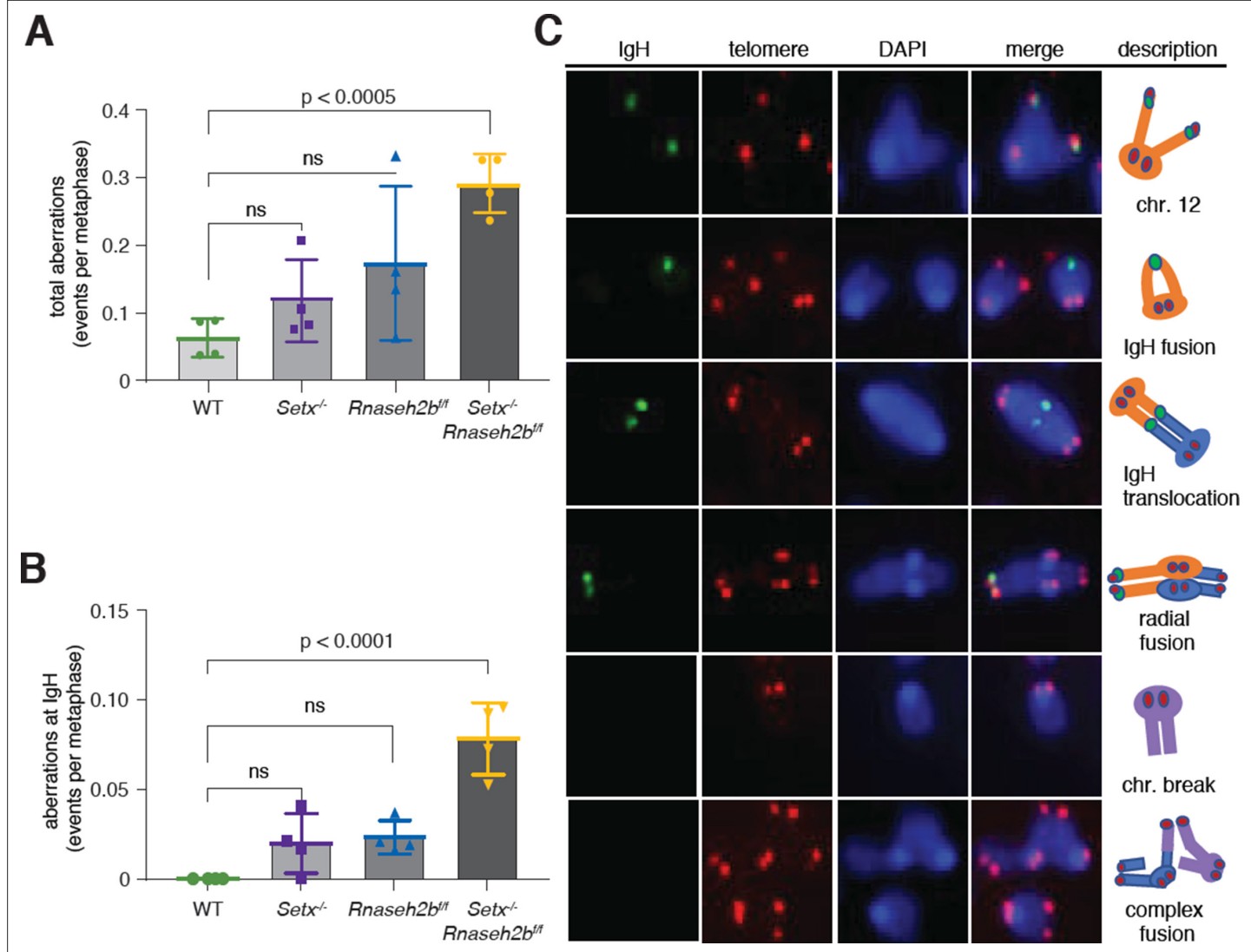

**Figure 3.** Increased IgH damage is observed in *Setx⁻/⁻Rnaseh2b^{f/f}* B cells. (**A**) Frequency of spontaneous DNA damage in WT, *Setx⁻/⁻*, *Rnaseh2b^{f/f}*, and *Setx⁻/⁻Rnaseh2b^{f/f}* cells. (**B**) Frequency of spontaneous DNA damage at IgH. (**C**) Representative images of the types of rearrangements produced. IgH-specific probe visualized in green, Telomere-specific probe visualized in red, DAPI is in blue. All cells were harvested 72 hr post-stimulation to IgG1 with LPS/IL-4/α-RP105. Error bars show standard deviation; statistical significance versus WT was determined by one-way ANOVA (n = 4 independent mice).

The online version of this article includes the following source data and figure supplement(s) for figure 3:

**Source data 1.** Numerical data used to generate graphs in *Figure 3A and B*.

**Figure supplement 1.** B cell development in the bone marrow is normal in cells lacking senataxin (Setx) and RNase H2B.

**Figure supplement 1—source data 1.** Numerical data used to generate graphs in *Figure 3—figure supplement 1B–D*.

**Figure supplement 2.** Ribonucleotide monophosphate (rNMP) incorporation and camptothecin (CPT) sensitivity in cells lacking senataxin (SETX) and RNase H2.

**Figure supplement 2—source data 1.** Uncropped alkaline gel image from WT, *Rnaseh2b^{f/f}*, *Setx⁻/⁻*, and *Setx⁻/⁻Rnaseh2b^{f/f}* cells.

**Figure supplement 2—source data 2.** Uncropped native gel image of WT, *Rnaseh2b^{f/f}*, *Setx⁻/⁻*, and *Setx⁻/⁻Rnaseh2b^{f/f}* B cells.

**Figure supplement 2—source data 3.** Numerical data used to generate graphs in *Figure 3—figure supplement 2E and F*.

**Figure supplement 3.** B cell proliferation and cell cycle in response to stimulation.

**Figure supplement 3—source data 1.** Numerical data used to generate graph in *Figure 3—figure supplement 3C*.

conditional gene deletion using *CD19^cre* leads to 75–80% deletion in developing pre-B cells in the BM (*Rickert et al., 1997*). To determine whether RNase H2 activity directly contributes to R loop metabolism at IgH during B cell activation, we expressed FLAG-tagged RNASEH2B in *Setx^-/-Rnaseh2b^f/f* splenic B cells by retroviral infection (*Figure 4A*). Re-expression of FLAG-tagged RNASEH2B suppressed rNMP mis-incorporation measured by alkaline gel electrophoresis, indicating the FLAG tag did not substantially interfere with RNase H2 complex formation or its ability to recognize and cleave rNMPs covalently attached to DNA (*Figure 4B and C*; *Chon et al., 2013*). We also found that RNASEH2B re-expression significantly reduced DNA:RNA hybrid signal at Sm to levels similar to *Setx^-/-* cells (*Figure 4D*). Finally, we also observed significant enrichment of RNASEH2B by ChIP at Sγ1 compared to Eµ, which does not exhibit high levels of DNA:RNA hybrids (*Figure 4E*; FLAG vs. IgG control). RNASEH2B was also consistently enriched at Sµ relative to IgG; however, this result was not significant due to inter-experiment variability (*Figure 4E*). Together, these results indicate that RNase H2 activity contributes to DNA:RNA hybrid removal at IgH during CSR. We were not able to measure SETX binding to IgH as commercially available antibodies did not IP murine SETX (data not shown) and the size of *Setx* precludes expression by retroviral infection as it encodes for a 2646 amino acid protein.

## Persistent IgH breaks and translocations are dependent on AID activity

To determine whether AID activity is required for the persistent DSBs observed at IgH, we next stimulated cells with α-RP105 alone. Stimulation with α-RP105 induces cell proliferation; however, AID expression is minimal compared to stimulation also containing LPS or LPS+IL-4 (*Figure 5A and B*; *Callén et al., 2007*). CSR to IgG1 was lower than 1% in all genotypes analyzed, correlating CSR efficiency with AID expression (*Figure 5C*). Total DNA damage levels were similar to LPS/IL-4/α-RP105-stimulated cells; however, we did not detect any DSBs or translocations at IgH in α-RP105-stimulated cells for any genotype (*Figure 5D* and *Supplementary file 1*). These results indicate that the IgH aberrations observed in *Setx^-/-Rnaseh2b^f/f* cells were AID-dependent. Further, only *Setx^-/-Rnaseh2b^f/f* cells consistently accumulate spontaneous unrepaired breaks in the absence of AID expression. It is possible that stimulation with α-RP105 alone alters transcriptional activity and R loop formation within IgH compared to stimulation with LPS/IL-4/α-RP105, altering the potential for R loop-induced IgH breaks. To confirm IgH damage was dependent on AID, we next generated *Aicda^-/-*, *Aicda^-/-Setx^-/-*, *Aicda^-/-Rnaseh2b^f/f*, and *Aicda^-/-Setx^-/-Rnaseh2b^f/f* mice, stimulated B cells with LPS/IL-4/α-RP105 for 72 hr, and performed IgH FISH on metaphase spreads. We observed no IgH breaks in LPS/IL-4/α-RP105-stimulated cells lacking AID; however, *Aicda^-/-Setx^-/-Rnaseh2b^f/f* B cells had elevated levels of non-IgH damage similar to *Setx^-/-Rnaseh2b^f/f* controls (*Figure 5E*). Further, ~7% of control *Setx^-/-Rnaseh2b^f/f* cells stimulated side-by-side had IgH breaks, consistent with previous experiments (*Figure 5F* and *Supplementary file 1*). Together, these results show that the persistent IgH breaks and translocations observed in *Setx^-/-Rnaseh2b^f/f* cells are AID-dependent.

## AID expression and recruitment to switch regions are not altered in SETX or RNase H2-deficient cells

AID overexpression induces high levels of DSBs, increasing CSR efficiency and unrepaired breaks at IgH. To determine whether loss of SETX or RNase H2 affected AID expression, we first measured *Aicda* transcript levels. We found that *Aicda* transcripts were similar to WT levels in *Setx^-/-*, *Rnaseh2b^f/f*, and *Setx^-/-Rnaseh2b^f/f* cells, indicating that AID gene regulation is not altered (*Figure 5—figure supplement 1A*). AID stability is also regulated during CSR. To determine whether protein levels were altered, we next measured protein abundance and found that AID expression was similar in all four genotypes (*Figure 5—figure supplement 1B and C*). These results show that the increased damage at IgH is not due to AID overexpression.

Enhanced recruitment of AID to switch regions positively correlates with DSB formation and CSR efficiency. To determine whether defective R loop removal alters AID recruitment to switch regions, we performed chromatin immunoprecipitation (ChIP) of AID in WT, *Setx^-/-*, *Rnaseh2b^f/f*, and *Setx^-/-Rnaseh2b^f/f* cells 60 hr post-stimulation. We found similar levels of AID recruitment at both the Sγ1 and Sµ switch regions in WT, *Setx^-/-*, *Rnaseh2b^f/f*, and *Setx^-/-Rnaseh2b^f/f* cells by ChIP-qPCR (*Figure 5—figure supplement 1D*). All four genotypes examined showed enrichment at Sγ1 and Sµ compared to *Aicda^-/-* cells. These results indicate that AID targeting to chromatin is not significantly altered in

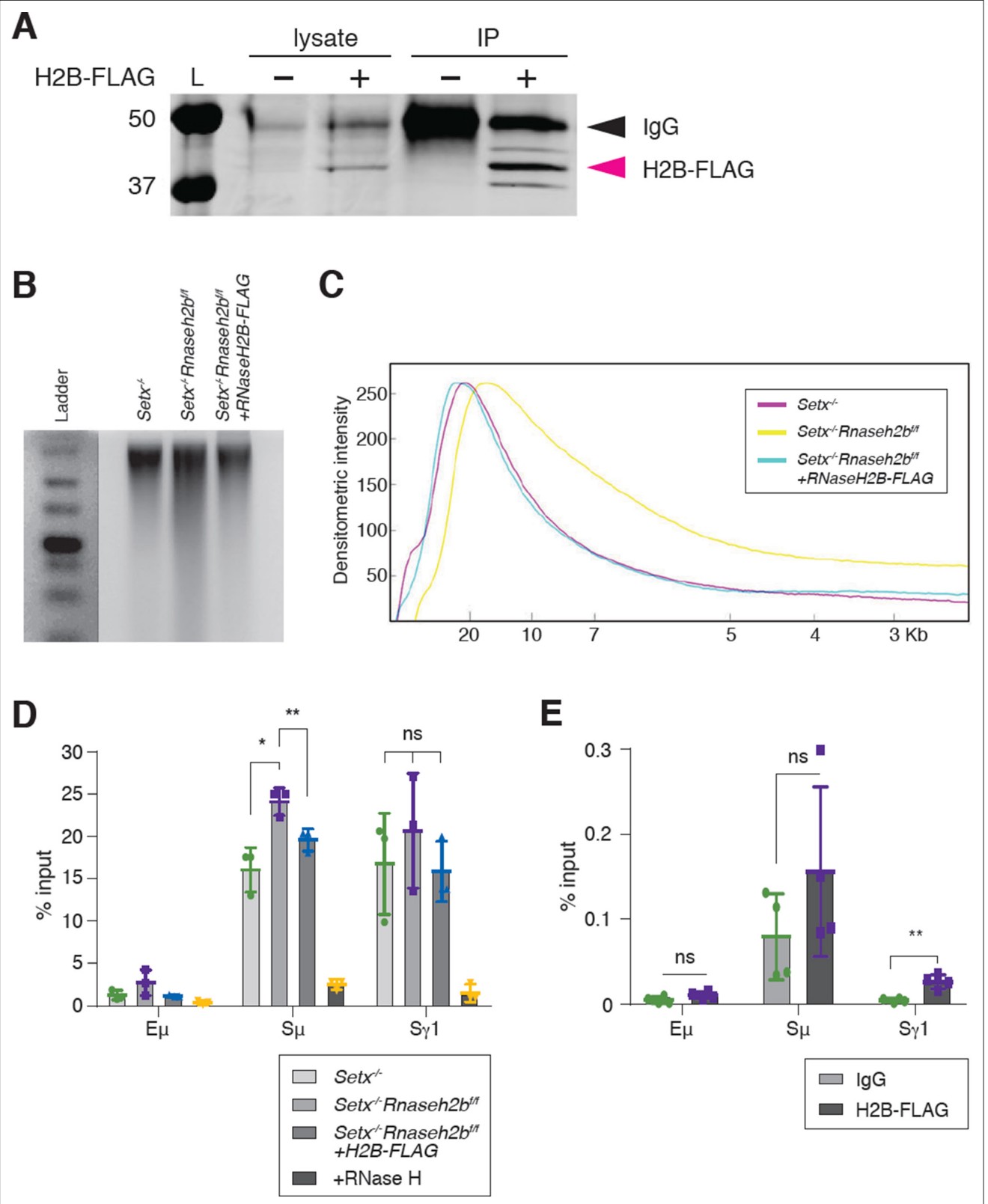

**Figure 4.** RNase H2 activity rescues DNA:RNA hybrid levels in stimulated B cells. (**A**) Total cell lysates were extracted from *Setx⁻ᐟ⁻Rnaseh2b^f/f* cells stimulated for 96 hr with LPS/IL-4/α-RP105. Cells were infected with empty vector or retrovirus, expressing FLAG-RNaseH2B, then subjected to immunoprecipitation and immunoblotting with indicated antibodies. (**B**) Representative image of alkaline gel from *Setx⁻ᐟ⁻, Setx⁻ᐟ⁻Rnaseh2b^f/f* EV, and *Setx⁻ᐟ⁻Rnaseh2b^f/f*+FLAG-RNaseH2B cells (n = 3 independent mice/genotype). (**C**) Densitometry trace of representative alkaline gel in (**B**). (**D**) DNA:RNA

*Figure 4 continued on next page*

*Figure 4 continued*

hybrid immunoprecipitation (DRIP) assay was performed with S9.6 antibody on *Setx*$^{-/-}$, *Setx*$^{-/-}$*Rnaseh2b*$^{f/f}$, and *Setx*$^{-/-}$*Rnaseh2b*$^{f/f}$+FLAG-RNaseH2B-expressing cells stimulated with LPS/IL-4/α-RP105 for 96 hr. RNase H treatment of *Setx*$^{-/-}$*Rnaseh2b*$^{f/f}$ sample was a negative control. Relative enrichment was calculated as chromatin immunoprecipitation (ChIP)/input, and the results were replicated in three independent experiments. Error bars show standard deviation; statistical analysis was performed using one-way ANOVA. (**E**) ChIP analysis for FLAG-RNaseH2B occupancy in Eμ, Sμ, and Sγ regions of primary B cells in response to LPS/IL-4/α-RP105 stimulation. Relative enrichment was calculated as ChIP/input. Error bars show standard deviation. Statistical analysis was performed using Student's *t*-test (n = 3 mice/genotype).

The online version of this article includes the following source data and figure supplement(s) for figure 4:

**Source data 1.** Uncropped western blot for RNaseH2B-FLAG expression in B cells under retroviral infection.

**Source data 2.** Uncropped alkaline gel from retrovirally infected cells.

**Source data 3.** Numerical data used to generate graphs in *Figure 4D and E*.

**Figure supplement 1.** Independent repeats *of FLAG*-RNaseH2B chromatin immunoprecipitation (ChIP).

**Figure supplement 1—source data 1.** Numerical data used to generate graphs in *Figure 4—figure supplement 1*.

*Setx*$^{-/-}$*Rnaseh2b*$^{f/f}$ cells, and enhanced AID targeting to switch regions is unlikely to be the cause of the increased IgH instability.

## RNA polymerase association with switch regions is normal in SETX- and RNase H2-deficient cells

AID physically associates with the transcription factor Spt5, leading to deamination both within and outside switch regions (*Pavri et al., 2010*; *Stanlie et al., 2012*). It is possible slower R loop turnover will increase the dwell time of RNA polymerase II (PolII) at R loop-forming genes. To determine whether PolII association at switch regions is increased, we performed ChIP of activated PolII (PolII-S5P) in all four genotypes. We found that WT, *Setx*$^{-/-}$, *Rnaseh2b*$^{f/f}$, and *Setx*$^{-/-}$*Rnaseh2b*$^{f/f}$ cells all had similar levels of PolII-S5P at both Sμ and Sγ1 switch regions (*Figure 5—figure supplement 1E*). PolII ChIPs showed variability particularly in *Rnaseh2b*$^{f/f}$ cells; however, IgH breaks consistently increased *Setx*$^{-/-}$*Rnaseh2b*$^{f/f}$ cells in every experiment. Thus, we conclude that increased PolII association at switch regions is not the cause of persistent IgH DNA damage observed in *Setx*$^{-/-}$*Rnaseh2b*$^{f/f}$ cells.

## Switch junctions show elevated insertions in *Setx*$^{-/-}$*Rnaseh2b*$^{f/f}$ cells

DSBs are generated by creating single-strand nicks on the template and non-template DNA strands, potentially creating staggered DSB ends. DSBs with limited overhangs (0–2 nucleotides) are candidates for classical NHEJ (cNHEJ), while breaks containing longer overhangs or mismatches may be repaired by alternative end-joining (alt-EJ). To determine how AID-induced breaks are repaired in SETX and RNase H2-deficient cells, we performed linear amplification-mediated high-throughput genome-wide translocation sequencing (LAM-HTGTS) (*Hu et al., 2016*). Within Sμ-Sγ1 junctions, we observed a significant decrease in blunt joins specifically in *Setx*$^{-/-}$*Rnaseh2b*$^{f/f}$ cells (27.3% in WT vs. 17.7% in double-deficient, p=0.031; *Figure 6A*). In contrast, insertions were significantly increased specifically in *Setx*$^{-/-}$*Rnaseh2b*$^{f/f}$ cells (21.1% in WT vs. 37.7% in double-deficient, p=0.008; *Figure 6A*). Insertion events cells were also longer in *Setx*$^{-/-}$*Rnaseh2b*$^{f/f}$ than those observed in WT cells, with significant increases in 3 and 4 or more bp insertions (*Figure 6—figure supplement 1C*; p=0.004 for 3 bp insertions, p=0.047 for 4 or more bp insertions). *Setx*$^{-/-}$ cells also showed increased insertions in Sμ-Sγ1 junctions; however, it was not significant (21.1% vs. 29.5%, p=0.059; *Figure 6A*). In addition, junctions with microhomology (MH) were modestly reduced in all mutants, with the largest reduction observed in *Setx*$^{-/-}$*Rnaseh2b*$^{f/f}$ cells (*Figure 6A*). From these results, we conclude that concomitant loss of SETX and RNase H2 affects repair pathway choice during CSR, increasing error-prone alt-EJ pathways that promote insertions while reducing blunt junctions. AID converts cytosines in switch regions to uracils in genomic DNA to initiate CSR (*Muramatsu et al., 2000*; *Revy et al., 2000*). If the resultant uracils are not recognized and removed by UNG, DNA polymerase erroneously incorporates an A in the complementary strand (*Rada et al., 2002*). It is possible excess R loops hinder uracil recognition and excision from DNA, resulting in increased transversion mutations. Indeed, we also observed a twofold increase in C>T transition mutations in *Setx*$^{-/-}$*Rnaseh2b*$^{f/f}$ cells (*Figure 6—figure supplement 1D*; p=0.048). G>A transition mutations were also 1.5-fold higher than WT cells; however, this was not significantly different (*Figure 6—figure supplement 1D*; p=0.105).

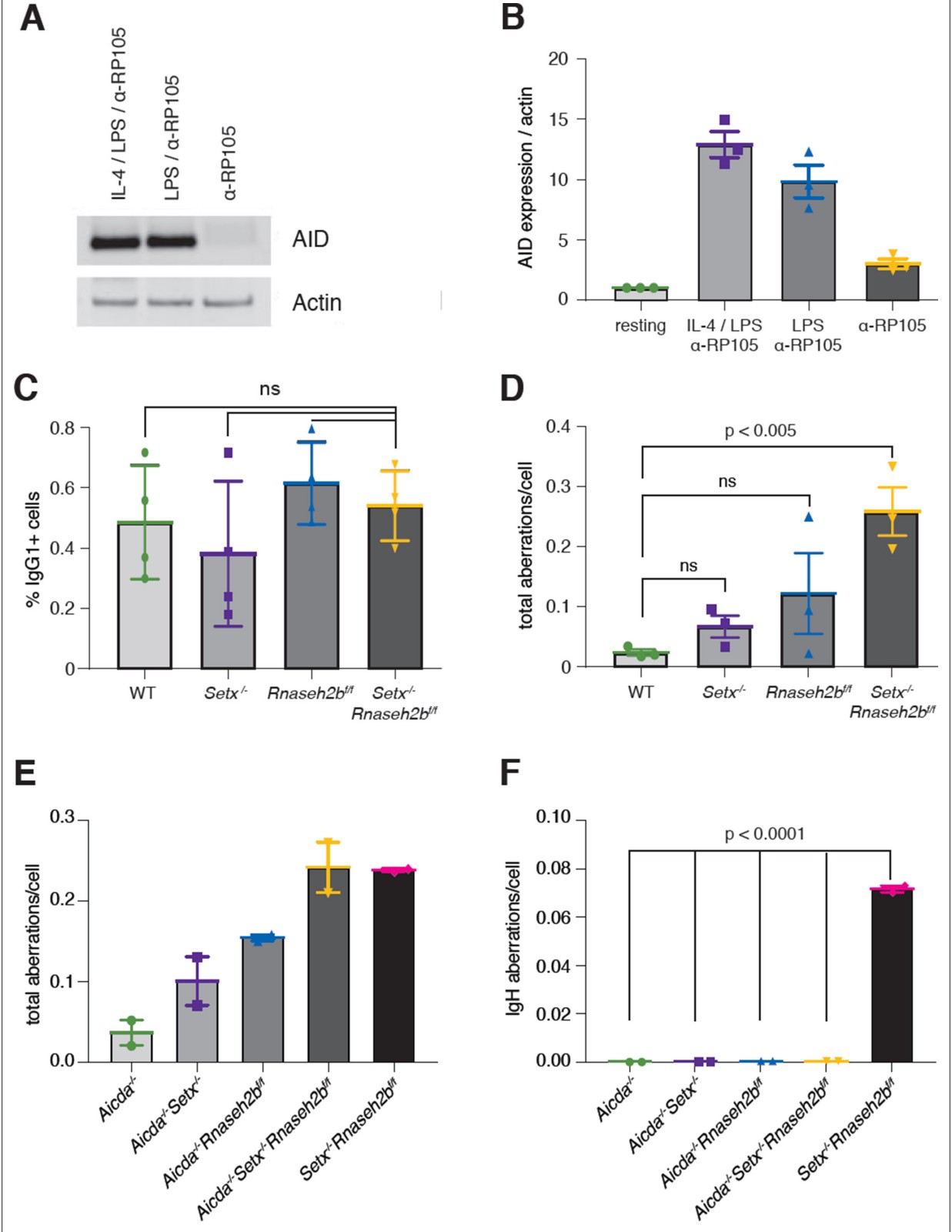

**Figure 5.** Activation-induced cytidine deaminase (AID) activity is required for persistent IgH breaks in *Setx⁻/⁻Rnaseh2b^{f/f}* B cells. (**A**) AID protein levels in WT B cells 72 hr post-stimulation to indicated isotypes (IgG1, LPS/IL-4/α-RP105; IgG3 with LPS/α-RP105; and α-RP105 alone). Actin served as a loading control. (**B**) Quantification of AID protein expression relative to Actin for three independent experiments, with AID expression in resting cells set as 1. Error bars show standard deviation. (**C**) Percent of cells undergoing class switch recombination (CSR) to IgG1 in B cells in response to α-RP105

*Figure 5 continued on next page*

*Figure 5 continued*

stimulation. Error bars show standard deviation; statistical significance between each genotype was determined by one-way ANOVA (n = 3 mice/genotype). (**D**) Frequency of total spontaneous DNA damage under anti-RP105 treatment in vitro. Error bars show the standard deviation; statistical significance versus WT was determined by one-way ANOVA (n = 3 mice/genotype). (**E**) Frequency of total spontaneous DNA damage 72 hr post-stimulation with LPS/IL-4/α-RP105 in *Aicda⁻/⁻*, *Aicda⁻/⁻Setx⁻/⁻*, *Aicda⁻/⁻Rnaseh2b^f/f*, *Aicda⁻/⁻Setx⁻/⁻Rnaseh2b^f/f*, and *Setx⁻/⁻Rnaseh2b^f/f* cells (n = 2 independent mice/genotype). (**F**) Frequency of spontaneous IgH damage in *Aicda⁻/⁻*, *Aicda⁻/⁻Setx⁻/⁻*, *Aicda⁻/⁻Rnaseh2b^f/f*, *Aicda⁻/⁻Setx⁻/⁻Rnaseh2b^f/f*, and *Setx⁻/⁻Rnaseh2b^f/f* cells 72 hr after stimulation with LPS/IL-4/α-RP105. Error bars show the standard deviation; statistical significance versus WT was determined by one-way ANOVA.

The online version of this article includes the following source data and figure supplement(s) for figure 5:

**Source data 1.** Uncropped Western blots of activation-induced cytidine deaminase (AID) and actin protein expression.

**Source data 2.** Numerical data used to generate graphs in *Figure 5B–F*.

**Figure supplement 1.** *Aicda* expression, enrichment, and recruitment in cells lacking senataxin (SETX) and RNase H2.

**Figure supplement 1—source data 1.** Activation-induced cytidine deaminase (AID) expression test.

**Figure supplement 1—source data 2.** Numerical data used to generate graphs in *Figure 5—figure supplement 1A, C–E*.

To further assess junction formation, we cloned and sequenced individual Sμ-Sγ1 switch junctions from all four genotypes. Here, *Setx⁻/⁻Rnaseh2b^f/f* cells showed no significant difference in MH use from WT cells. This is not surprising as the modest reduction observed in MH use by HTGTS (from 50% to 45%) would be challenging to detect by cloning individual junctions. Similar to HTGTS analyses, *Setx⁻/⁻Rnaseh2b^f/f* and *Setx⁻/⁻* cells had a significantly increased level of insertions at junction sites compared to WT cells (*Figure 6B and C*)—where insertions are defined as junctions containing nucleotides that did not map to either Sμ or Sγ1 (*Figure 6B*) (WT vs. *Setx⁻/⁻* $6.19 \times 10^{-7}$, WT vs. *Setx⁻/⁻Rnaseh2b^f/f*: $1.25 \times 10^{-5}$; chi-square goodness-of-fit test). Overall, these results support the HTGTS analyses showing an increase in insertion events in *Setx⁻/⁻Rnaseh2b^f/f* cells indicating an increase in alt-EJ.

## *Setx⁻/⁻Rnaseh2b^f/f* cells have reduced KU70/80 binding

The reduction of DNA repair events resulting in blunt Sμ-Sγ1 junctions suggests that cNHEJ is reduced in *Setx⁻/⁻Rnaseh2b^f/f* cells. The Ku70/80 heterodimer is a key initial step of cNHEJ that binds DNA ends and recruiting ligase 4 to covalently reconnect the broken DNA ends (*Nick McElhinny et al., 2000*). B cells lacking Ku70 or Ku80 exhibit reduced CSR efficiency, and the resulting junctions exhibit a reduced frequency of blunt junctions and a concomitant increase in the frequency of junctions harboring MH and insertions (*Boboila et al., 2010*; *Casellas et al., 1998*; *Guirouilh-Barbat et al., 2007*; *Manis et al., 1998*). To determine whether Ku70/80 recruitment to switch junctions is reduced, we performed ChIP of the Ku70/80 heterodimer in WT, *Setx⁻/⁻*, *Rnaseh2b^f/f*, and *Setx⁻/⁻Rnaseh2b^f/f* cells 60 hr post-stimulation (*Figure 6D*). Ku70/80 competes with the homologous recombination protein Rad52 at switch regions during CSR and contributes to Ku-independent repair (*Zan et al., 2017*). To assess whether Rad52 binding is increased, we next measured Rad52 recruitment to switch regions by ChIP. We observed an increase in Rad52 binding; however, the increase was not statistically significant (*Figure 6E*). From these results, we conclude the reduction in blunt joins indeed results from reduced Ku70/80 binding, and ends are repaired by alternative mechanisms.

## Discussion

In this work, we used combined deletion of *Setx* and *Rnaseh2b* to investigate the role R loops play in CSR. Co-transcriptional R loop formation has emerged as a regulator of CSR involved in targeting AID to the appropriate switch regions for DSB formation. AID recruitment to chromatin is a highly regulated act as off-target AID activity promotes IgH and non-IgH DSBs and translocations associated with carcinogenesis (*Ramiro et al., 2004*; *Robbiani et al., 2008*; *Robbiani et al., 2009*). However, persistent R loops are also sources of replication-associated DNA damage and genome instability (*Crossley et al., 2019*; *Marnef and Legube, 2021*; *Prado and Aguilera, 2005*; *Stork et al., 2016*). Here, we show that combined loss of SETX and RNase H2 leads to an increase in switch region R loop abundance and IgH damage in the form of unrepaired DSBs and translocations in mitotic cells. While SETX and RNase H2 also have independent functions in DNA repair, loss of either factor alone was not sufficient to significantly increase either R loops or IgH instability. The increase in unrepaired breaks

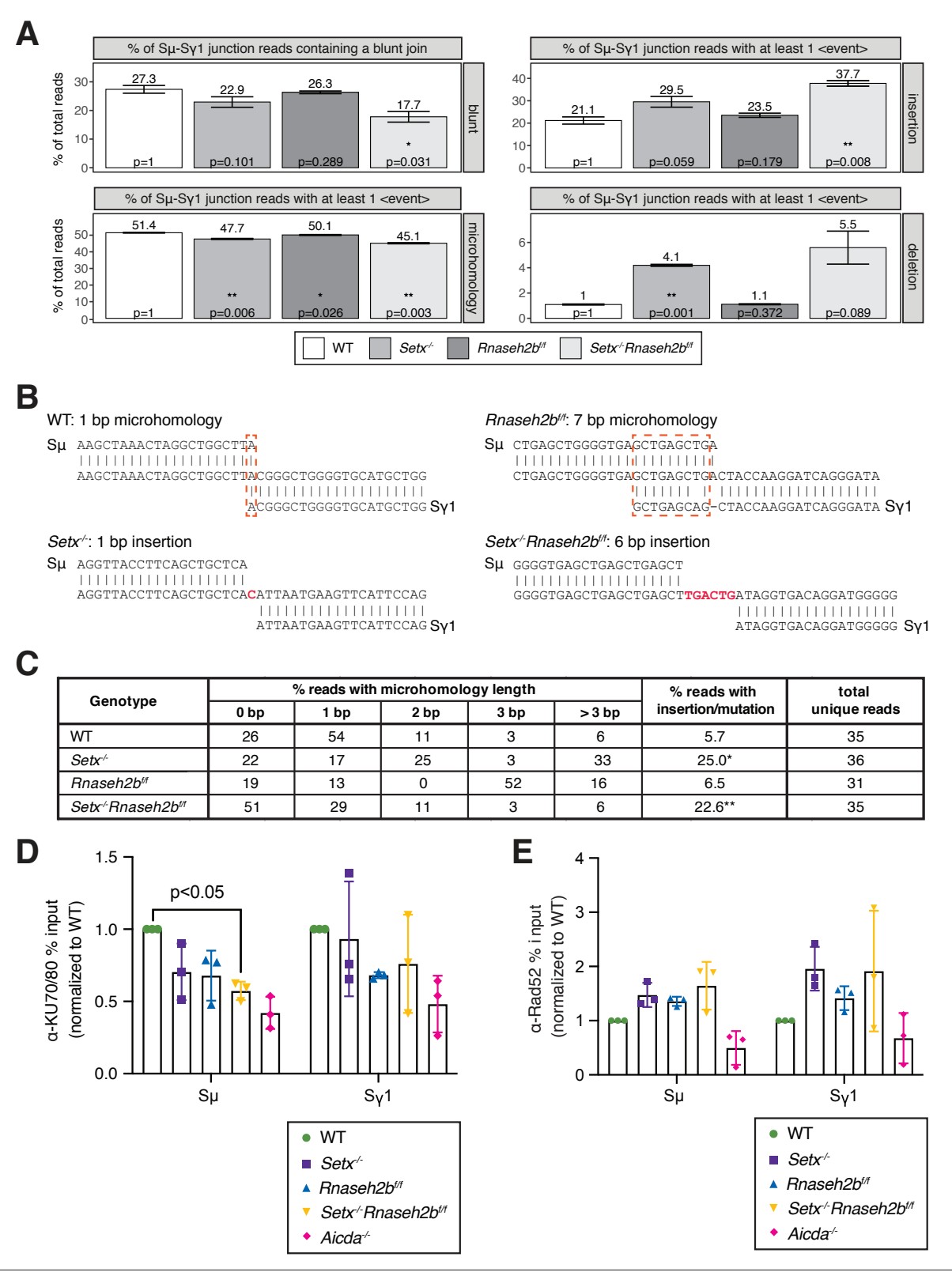

**Figure 6.** Altered switch junctions in in *Setx⁻ᐟ⁻*, *Rnaseh2bᶠ/ᶠ*, and *Setx⁻ᐟ⁻Rnaseh2bᶠ/ᶠ* B cells. (**A**) Linear amplification-mediated high-throughput genome-wide translocation sequencing (LAM-HTGTS) analysis showing the percentage of sequenced junction events harboring blunt joins, microhomology (MH) use, insertions, or deletions at junction sites. LAM-HTGTS uses two technical replicates from genomic DNA isolated from cells 72 hr after LPS/IL-4/α-RP105 stimulation. p-Values are calculated using Student's *t*-test. Sequencing reads may harbor more than one event class (insertion, deletion,

*Figure 6 continued on next page*

*Figure 6 continued*

mutation, MH) with the resulting junctions having a more complex result; a junction exhibiting MH may also have a deletion or mismatches in flanking DNA. Thus, junction types were quantified on whether they contained at least one event of the class listed in the side bar (deletions, insertions, MH). (**B**) Representative nucleotide sequences surrounding representative Sμ-Sγ junctions from WT, *Setx⁻/⁻*, *Rnaseh2b^f/f*, and *Setx⁻/⁻Rnaseh2b^f/f* B cells from Sanger sequencing of cloned junctions. Overlap was determined by identifying the longest region at the switch junction of perfect uninterrupted donor/acceptor identity. Sμ and Sγ1 germline sequences are shown above and below each junction sequence, respectively. Regions of MH at junctions are boxed with a dashed red line, and insertions are in red bold text. Genomic DNA from sequencing experiments was isolated from two independent mice for each genotype. (**C**) Table with absolute numbers of uniquely mapping cloned switch junctions harboring MH and insertions in WT, *Setx⁻/⁻*, *Rnaseh2b^f/f*, and *Setx⁻/⁻Rnaseh2b^f/f* B cells 72 hr after stimulation with LPS/IL-4/α-RP105. p-Values were calculated using the chi-square goodness-of-fit test: *WT vs. *Setx⁻/⁻* $6.19 \times 10^{-7}$, ** WT vs. *Setx⁻/⁻Rnaseh2b^f/f*: $1.25 \times 10^{-5}$. (**D**) Chromatin immunoprecipitation (ChIP) analysis for KU70/KU80 occupancy in Sμ and Sγ regions of primary B cells in response to LPS/IL-4/α-RP105 stimulation. Relative enrichment was calculated as fold change relative to WT set to 1; error bars show standard deviation (n = 3 mice/genotype), statistical analysis versus WT was performed using Student's *t*-test. (**E**) ChIP analysis for RAD52 occupancy in Sμ and Sγ regions of primary B cells in response to LPS/IL-4/α-RP105 stimulation. Relative enrichment was calculated as fold change relative to WT set to 1; error bars show standard deviation (n = 3 mice/genotype).

The online version of this article includes the following source data and figure supplement(s) for figure 6:

**Source data 1.** Numerical data used to generate graphs in *Figure 6D and E*.

**Figure supplement 1.** Features of Sμ-Sγ1 junction sites.

was not accompanied by significant alterations in transcriptional activity, splicing, or AID recruitment to switch regions, indicating that persistent R loops do not impact AID recruitment or deamination. Rather, C>T mutations were increased in *Setx⁻/⁻Rnaseh2b^f/f* cells. Analysis of switch junctions showed an increase in insertion events in *Setx⁻/⁻Rnaseh2b^f/f* cells, indicating that DSB repair by mutagenic alt-EJ was enhanced. Switch junction analysis by Sanger sequencing and LAM-HTGTS revealed that loss of Setx alone increased the frequency of insertion, indicating that these assays are more sensitive than DRIP and ChIP experiments. However, Setx may influence DSB repair in other ways; it is implicated in recruiting Brca1 to transcriptional pause sites (*Hatchi et al., 2015*). From this work, we propose the shared function of SETX and RNase H2 in R loop removal promotes efficient CSR suppresses genome instability during CSR by stimulating efficient NHEJ.

## Separation between CSR efficiency and IgH instability

In contrast to a prior report describing a modest reduction in CSR in SETX-deficient cells, we found that CSR efficiency to IgG1, IgG2B, and IgA in *Setx⁻/⁻* and *Setx⁻/⁻Rnaseh2b^f/f* cells was comparable to WT cells, indicating that the majority of DSBs created productive junctions leading to cell surface expression (*Kazadi et al., 2020*). This is consistent with our results, which showed no change in germ-line transcription, RNAP2 association, or AID recruitment. Thus, why do *Setx⁻/⁻Rnaseh2b^f/f* cells have substantially increased IgH breaks without a detectable reduction in CSR? To date, all mouse models exhibiting increased IgH breaks by metaphase spread analysis also exhibit altered CSR efficiency: mice lacking mismatch repair (MMR) factors (MLH1, PMS2, Mbd4), DSB processing factors (CtIP, Exo1), NHEJ proteins (Ku70, Ku80, Xrcc4, Lig4), mediators (53bp1, Rnf8), or kinases (ATM, DNAPKcs) (*Alt et al., 2013*; *Bardwell et al., 2004*; *Boboila et al., 2010*; *Callén et al., 2007*; *Casellas et al., 1998*; *Franco et al., 2008*; *Grigera et al., 2017*; *Gu et al., 1997*; *Lee-Theilen et al., 2011*; *Li et al., 2010*; *Lumsden et al., 2004*; *Manis et al., 2002*; *Reina-San-Martin et al., 2004*; *Santos et al., 2010*; *Schrader et al., 2002*; *Ward et al., 2004*; *Yan et al., 2007*). Alternatively, enhanced AID expression or nuclear localization increases AID-associated damage; however, these elevate CSR frequency (*Robbiani et al., 2009*; *Uchimura et al., 2011*). Though concomitant loss of Setx and RNase H2B increased R loops at IgH and enhanced genome stability at IgH, it did not impact CSR frequency to IgG1. This apparent discrepancy may be due to the observation that R loops are increased specifically at Sμ but not Sγ1. DSB formation at Sγ1 are limiting for CSR; therefore, an R loop-mediated increase AID activity and DSB formation at Sμ is not likely to increase CSR frequency. Similarly, the reduction in Ku70/80 binding specifically at Sμ is also unlikely to have a major impact on CSR frequency (*Figure 6D*). Further, R loops at Sμ are more reliant on Setx and RNase H2 for their removal to create DSB ends appropriate for NHEJ (*Figure 1D*). Why the R loops at Sμ disproportionately require Setx and RNase H2 for R loop removal in comparison to the R loops formed at Sg1 is unclear. One possibility is that the R loops at Sμ are present in resting B cells, and Setx and RNase H2 may be recruited in G0 prior to stimulation. Sμ also has a higher level of DRIP signal than Sg1; thus, Sμ R loops form in a higher percentage of cells,

have a longer half-life, or both. Finally, RNAP2 recruitment to Sg1 requires additional transcription factors stimulated specifically by IL-4 such as Stat6 (*Linehan et al., 1998*). Therefore, it is possible that distinct transcription factors or other chromatin-modifying enzymes involved in switch region transcription induce differential Setx and/or RNase H2 recruitment to distinct switch regions.

Alt-EJ has been proposed to be a default pathway used when cNHEJ proteins are absent (*Bennardo et al., 2008*; *Boboila et al., 2010*; *Nussenzweig and Nussenzweig, 2007*; *Soulas-Sprauel et al., 2007*; *Stavnezer and Schrader, 2014*; *Yan et al., 2007*); however, we report a scenario where unrepaired breaks and mutations at switch junctions are increased without a concomitant reduction in CSR, suggesting that error-prone EJ becomes preferred even when all core cNHEJ factors are present. Indeed, repair by alternate pathways may contribute to why we observe not major defect in CSR though Ku70/80 binding is reduced in *Setx⁻/⁻Rnaseh2b^{f/f}* cells. This is not without precedent as cells lacking the nuclease Artemis exhibit increased MH use at switch junctions without substantially affecting CSR to most isotypes (*Du et al., 2008*; *Rivera-Munoz et al., 2009*). Taken together, our results indicate that DSB formation and DNA end-joining processes are largely intact in *Setx⁻/⁻Rnaseh2b^{f/f}* cells; however, blunt joins are reduced indicating that DSB repair pathway choice and cNHEJ efficiency are likely impaired.

## DNA structure and AID recruitment

The increase in C>T and G>A point mutations in *Setx⁻/⁻Rnaseh2b^{f/f}* cells may support a role for R loops in AID recruitment. In the absence of efficient R loop removal, AID may be recruited multiple times to the same hybrid-forming region; this scenario could lead to more deamination events without significantly altering AID association with chromatin as measured by ChIP. Crystal structures of AID revealed two nucleotide binding regions—the substrate channel itself and an assistant patch—indicating a preference for branched substrates (*Qiao et al., 2017*). This raises the possibility that other branched nucleotides could also be AID substrates. Indeed, an RNA-DNA fusion molecule can also bind and be deaminated by AID, raising the notion that R loop 'tails' also promote AID recruitment to switch regions (*Liu et al., 2022*). These structures can work independently or together to enrich AID association along switch regions (*Figure 7*). Of note, R loop tails are flexible and may cause AID association with template or non-template strands depending on ssDNA availability. R loops in switch regions can also promote the formation of G-quadruplexes in the non-template strand, a structure AID strongly binds (*Lim and Hohng, 2020*; *Qiao et al., 2017*). Switch RNAs themselves form G-quadruplexes and AID shows equal binding affinity for RNA and DNA G-quadruplexes (*Qiao et al., 2017*; *Zheng et al., 2015*). Thus, R loop stabilization within switch regions may promote the formation of DNA and RNA G-quadruplexes, as well as branched RNA-DNA substrates—all strong substrates for AID binding and activity. However, R loops present a double-edged sword; while their formation may promote AID recruitment and stimulate CSR, their sustained presence potentially perturbs uracil processing, resulting in an increase in C>T and G>A transition mutations during DNA replication.

## Efficient R loop removal promotes cNHEJ and genome integrity during CSR

We propose that R loops at switch regions initially promote AID activity; however, their persistence after break formation subsequently interferes with DSB end processing and/or joining, resulting in the IgH instability observed in *Setx⁻/⁻Rnaseh2b^{f/f}* cells (*Figure 3B and C*). This places SETX and RNase H2 downstream of DDX1, an RNA helicase, which promotes R loop formation and AID targeting by unwinding G4 quadruplex structures in switch transcripts (*Ribeiro de Almeida et al., 2018*). Normally, SETX and RNase H2 remove switch R loops along the non-template strand, promoting the formation of DSB ends appropriate for cNHEJ (*Figure 7*). In the absence of SETX and RNase H2, persistent switch region R loops potentially affects CSR in two distinct ways—by changing the type of DSBs created or by interfering with DSB repair protein recruitment.

In the first possibility, persistent R loops could block AID access to the template strand, potentially increasing the length of 5′ ssDNA tails at DSB ends. This model is supported by a potential role of the RNA exosome in CSR where AID association with RNA exosome components promotes deamination on both template and non-template DNA strands, presumably by removing RNA annealed to the template strand (*Basu et al., 2011*). However, we found no difference in AID recruitment by ChIP to support this; instead, our observed increase in C>T mutations in switch junctions is an indication

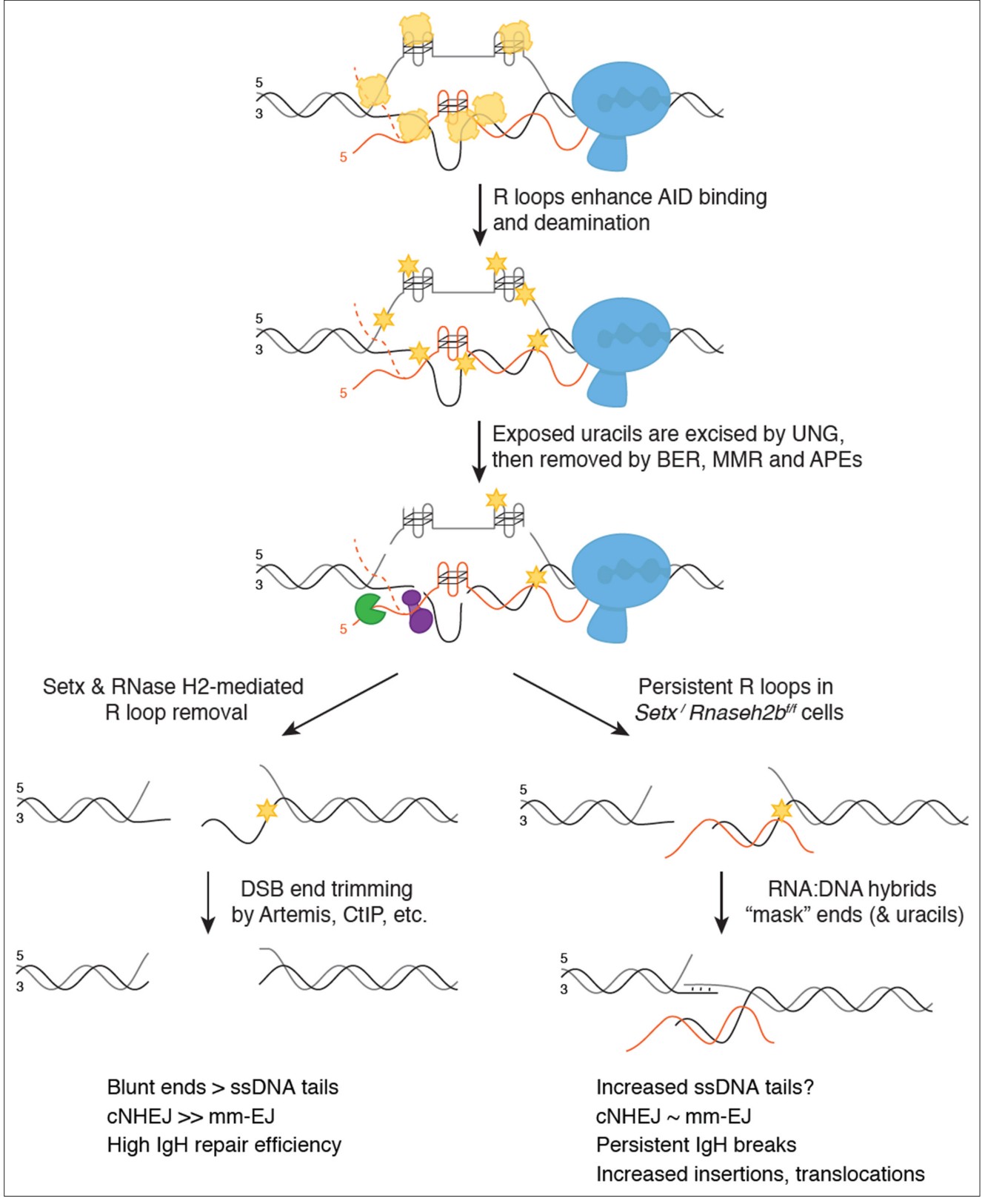

**Figure 7.** Model for senataxin (SETX) and RNase H2 in promoting efficient class switch recombination (CSR). When B cells are stimulated to undergo class switching, PolII-mediated transcription opens duplex DNA at recombining S regions. R loops formed during transcription promote activation-induced cytidine deaminase (AID) binding to ssDNA on the non-template strand first. SETX and RNase H2 then cooperate to remove switch region R loops, exposing ssDNA on the template strand for AID to bind. Extensive AID activity and uracil removal on both strands results in the formation of

*Figure 7 continued on next page*

*Figure 7 continued*

double-stranded breaks (DSBs) with limited single-strand DNA (ssDNA) overhangs at break ends ('blunted' ends), which are predominantly repaired by classical non-homologous end joining (cNHEJ). When SETX and RNase H2B are absent, R loops forming at S region are not efficiently removed and error-prone EJ is increased. This persistent R loop/RNA:DNA hybrid may affect CSR in two ways. One possible mechanism for increased error-prone EJ is that persistent R loops reduce the extent of ssDNA available for AID binding specifically on the template strand, increasing ssDNA tail length at DSB ends. Alternatively, persistent RNA:DNA hybrids may alter DNA repair protein recruitment to DSB ends, impeding end processing and/or ligation. Both possibilities reduce NHEJ efficiency, but do not affect overall CSR levels as the majority of breaks with long ssDNA tails are repaired by error-prone EJ. However, a subset of breaks are not repaired, leading to persistent DSBs that manifest as chromosome breaks and translocations in mitotic spreads.

of robust AID-mediated deamination. Persistent R loops could also interfere with the recognition and removal of AID-induced uracils, decreasing DSB formation and/or generating DSB ends with 3' or 5' ssDNA tails not immediately suitable for cNHEJ. Yet *Setx⁻/⁻Rnaseh2b^f/f* cells are proficient for CSR, demonstrating that processing of deaminated bases successfully produces switch region DSBs visible as persistent IgH breaks in metaphase chromosome spreads. Taken together, we conclude that elevated R loops caused by SETX and RNase H2 loss do not dramatically impede AID recruitment or DSB formation, though DSB end structure may be altered.

Efficient R loop removal may also be necessary after DSB formation. RNA:DNA hybrids at one or both DSB ends may prevent core NHEJ factors from binding. Indeed, we observed a reduction of Ku70/80 heterodimer association by ChIP at Sμ (*Figure 6D*). DSBs created during CSR often have staggered ends, requiring further processing for repair by cNHEJ (*Stavnezer and Schrader, 2014*). Persistent RNA:DNA hybrids at or near DSB ends could slow or block cNHEJ-mediated repair by protecting ssDNA overhangs from processing or by impeding cNHEJ protein binding to DSB ends. Most DSB ends can undergo trimming yielding substrates with short <3 bp overhangs, suitable for cNHEJ. Ku70/Ku80 has lyase activity, preferentially removing apurinic/apyridimic (AP) nucleotides from the 5' end of DSBs (5'-dRP) (*Roberts et al., 2010*). This activity appears restricted to short 5' overhangs, potentially lessening its interference with HR-mediated repair requiring long 3' ssDNA tails (*Strande et al., 2012*; *Symington, 2016*). MH-mediated end joining also requires trimming of ssDNA tails prior to ligation (*Chang et al., 2016*; *So and Martin, 2019*). In support of this notion, loss of Artemis increases the frequency of unrepaired IgH breaks and MH use at switch junctions (*Chang et al., 2016*; *Chang et al., 2017*; *Du et al., 2008*; *Franco et al., 2008*; *Rivera-Munoz et al., 2009*). Indeed, we found that MH use was reduced 6.3% in *Setx⁻/⁻Rnaseh2b^f/f* cells by LAM-HTGTS (*Figure 6A*). Thus, increased R loops in *Setx⁻/⁻Rnaseh2b^f/f* cells may impede end processing during alt-EJ.

R loop removal also appears important for later steps of HR, thus a role in NHEJ is not unexpected. Reducing R loop removal by SETX depletion does not impair DSB end resection but does slow the recruitment of the strand invasion factor Rad51 (*Cohen et al., 2018*). Thus, how persistent R loops influence DNA repair likely depends on substrate binding affinity of repair proteins. Early acting sensors as well as helicases and nucleases involved in resection may have evolved to recognize DNA substrates bound by RNA, while proteins catalyzing later steps such as filament formation and D loop formation may not.

It is interesting to note the consistently lower read count obtained from *Setx⁻/⁻Rnaseh2b^f/f* B samples. Increased R loop stability may interfere with PCR extension steps as genomic DNA is extracted without RNase H treatment. Alternatively, the low read count may result from poor PCR amplification of junction events involving sequences close to one or both nested primers, mutations within primer binding sites, or complex repair events that included insertion of de novo sequence. Indeed, junctions isolated from *Setx⁻/⁻Rnaseh2b^f/f* cells exhibit a high number of insertion events (*Figure 6A* and *Figure 6—figure supplement 1C*). Error-prone polymerases such as Pol theta and Pol zeta perform gap-filling around the annealed region in MH-mediated EJ, indicating a role for these enzymes in CSR (*Mateos-Gomez et al., 2015*; *Schenten et al., 2009*; *Yu and McVey, 2010*). Indeed, switch junctions formed in Pol theta-deficient cells notably lack insertions >1 bp, indicating that it is required for these events (*Yousefzadeh et al., 2014*). Thus, R loops may impact the recruitment of specific error-prone polymerases to DSB ends during CSR either directly or indirectly. Further investigation of repair protein recruitment and repair kinetics in *Setx⁻/⁻*, *Rnaseh2b^f/f*, and *Setx⁻/⁻Rnaseh2b^f/f* cells will help delineate how persistent R loops influence DSB end processing and repair pathway choice. Finally, RNA transcripts can also act as templates for recombinational repair-mediated Rad52 or MH-mediated EJ

mediated by Pol theta (*Keskin et al., 2014*; *Mazina et al., 2017*; *McDevitt et al., 2018*; *Storici et al., 2007*). Thus, RNAs arising from the switch regions themselves or other transcribed regions may act as templates for the insertions observed at switch junctions.

These two models of how persistent R loops impact CSR are not mutually exclusive, as DSB end structure directly affects repair protein binding affinity and which proteins are necessary for successful repair (*Chang et al., 2017*; *Serrano-Benítez et al., 2019*; *Symington, 2016*). We propose that persistent R loops promote the formation of long (>6 bp) ssDNA tails, increasing the frequency of error-prone EJ at switch joins in *Setx⁻/⁻*, *Rnaseh2b^f/f^*, and *Setx⁻/⁻Rnaseh2b^f/f^* cells. In the absence of both SETX and RNase H2, RNA:DNA hybrids persist at a subset of DSB ends, interfering with efficient repair by cNHEJ and leading to persistent IgH breaks and translocations observed in mitosis. However, many enzymes—including the RNA exosome, RNase H1, and additional RNA:DNA-specific helicases—can also remove these structures, indicating that the resolution step of CSR is slowed but not blocked. Indeed, SETX is not unique in its ability to unwind RNA:DNA hybrids; therefore, additional R loop resolving enzymes may also influence CSR. Multiple helicases have been implicated in R loop removal associated with replication stress, including Aquarius, DDX19, and DDX21, among others (*Hodroj et al., 2017*; *Sollier et al., 2014*; *Song et al., 2017*). It will be interesting to determine whether these enzymes exhibit similar or distinct effects on CSR.

# Materials and methods

## Key resources table

| Reagent type (species) or resource | Designation | Source or reference | Identifiers | Additional information |
|---|---|---|---|---|
| Genetic reagent (*Mus musculus*) | *Rnaseh2b^f/f^* | PMID:2802351 | RRID:MGI:5911393 | Dr. Axel Roers (University of Technology Dresden) |
| Genetic reagent (*M. musculus*) | *Setx⁻/⁻* | PMID:23593030 | RRID:MGI:5697060 | Dr. Martin F Lavin (Queensland Institute of Medical Research) |
| Genetic reagent (*M. musculus*) | *Cd19^cre^* | PMID:9092650 | RRID:MGI:5614310 | Dr. Klaus Rajewsky (University of Cologne) |
| Genetic reagent (*M. musculus*) | *Aicda⁻/⁻* | PMID:11007474 | RRID:MGI:2156156 | |
| Recombinant DNA reagent | Migr1-RNaseH2B-FLAG | This paper | | N-terminally FLAG tagged mouse Rnaseh2b |
| Antibody | Anti-S9.6 (mouse monoclonal) | Chedin lab | | DRIP (1:500) Dot blot (1:1000) |
| Antibody | IgG (H+L) Cross-Adsorbed Secondary Antibody, Alexa Fluor 680 (goat anti-mouse polyclonal) | Thermo Fisher | Cat# A21057 | Dot blot (1:5000) |
| Antibody | APC CD19 (rat anti-mouse monoclonal) | BD Pharmingen | Cat# 561738 | Flow cytometry (1:100) |
| Antibody | PE-Cy7-CD43 (rat anti-mouse monoclonal) | BD Pharmingen | Cat# 562866 | Flow cytometry (1:20) |
| Antibody | PE -IgM (rat anti-mouse monoclonal) | BD Pharmingen | Cat# 562033 | Flow cytometry (1:100) |
| Antibody | BV421-CD11b (rat anti-mouse monoclonal) | BD Pharmingen | Cat# 562605 | Flow cytometry (1:100) |
| Antibody | PerCP- CD45R/B220 (rat anti-mouse monoclonal) | BD Pharmingen | Cat# 553093 | Flow cytometry (1:100) |
| Antibody | Anti-AID (mouse monoclonal) | Thermo Fisher | Cat# 39-2500 | CHIP (1:200) WB (1:500) |
| Antibody | Anti-KU70/KU80 (mouse monoclonal) | Invitrogen | Cat# MA1-21818 | CHIP (1:100) |
| Antibody | Anti-Rad52 (rabbit polyclonal) | ABclonal | Cat# A3077 | CHIP (1:500) |
| Antibody | Anti-RNA polymerase II (mouse monoclonal) | Abcam | Cat# AB5408 | CHIP (1:500) |
| Antibody | IgG- Isotype Control (mouse polyclonal) | Abcam | Cat# AB37355 | CHIP (1:500) |
| Antibody | Anti-CD180 (rat anti-mouse monoclonal) | BD Pharmingen | Cat# 552128 | CSR (1:2000) |

*Continued on next page*

*Continued*

| Reagent type (species) or resource | Designation | Source or reference | Identifiers | Additional information |
|---|---|---|---|---|
| Peptide, recombinant protein | Recombinant murine IL-4 | PeproTech | Cat# 214-14 | CSR (1:2000) |
| Peptide, recombinant protein | Lipopolysaccharides | MilliporeSigma | Cat# L2630 | CSR (1:2000) |
| Commercial assay or kit | Beckman Coulter AMPURE XP | Beckman Coulter | Cat# A63881 | Size selection |
| Commercial assay or kit | SPHERO AccuCount Blank Particles | Spherotech | Cat# ACBP-50-10 | Flow cytometry |

## Mice

$Setx^{-/-}$, $Rnaseh2b^{f/f}$, $Aicda^{-/-}$, and $CD19^{cre}$ mice were previously described and used to generate $Setx^{-/-}$ $Rnaseh2b^{f/f}$ $CD19^{cre}$ and $Aicda^{-/-}$ $Setx^{-/-}$ $Rnaseh2b^{f/f}$ $CD19^{cre}$ mice (*Becherel et al., 2013*; *Hiller et al., 2012*; *Muramatsu et al., 2000*; *Rickert et al., 1997*). Both $Setx^{-/-}$ $Rnaseh2b^{f/f}$ and $Aicda^{-/-}$ $Setx^{-/-}$ $Rnaseh2b^{f/f}$ $CD19^{cre}$ mice were a mixed cross of C57BL6/129Sv ($Setx^{-/-}$) and C57BL/6 ($Aicda^{-/-}$ and $Rnaseh2b^{f/f}$ $CD19^{cre}$). For CSR to IgG1, we used a minimum of 10 mice per genotype to reach a power of 0.8 with an expectation to see a 25% difference. A priori power calculations to determine mouse numbers were based on reported CSR data from *Kazadi et al., 2020* and performed using G*power 3.1. Since no difference in CSR was detected for IgG1, subsequent CSR studies were based on similar reports in the literature, using a minimum of n = 4 mice per genotype. For molecular analyses, we used n = 3 mice to reach a power of 80% for p-value calculations estimating differences to be consistently ≥50%. Primary cells from both male and female mice were used to eliminate sex bias. No difference was observed between sexes. The age of mice was matched in experiments to reduce bias as this variable is known to alter DNA repair efficiency and CSR. No randomization or blinding was used in mouse experiments. All mouse experiments were performed in accordance with the protocols approved by the UC Davis Institutional Animal Care and Use Committee (IACUC protocol #20042).

## B cell stimulation

$CD43^-$ resting B cells were isolated using the Dynabeads untouched CD43 mouse B cell isolation kit (Thermo Fisher, 11422D). Isolated B cells were cultured in B cell media (BCM, RPMI-1640 supplemented with 10% fetal calf serum, 1% L-glutamine, 50 IU/ml penicillin/streptomycin, 1% sodium pyruvate, 53 μM 2-mercaptoethanol, 10 mM HEPES). B cells were stimulated with LPS, α-RP105, and interleukin 4 (IL-4) for IgG1, LPS/α-RP105/TGF-B for IgG2b and LPS/α-RP105/TGF-B/CD40L for IgA CSR.

## DRIP analyses (DRIP, DRIP-seq, dot blot, qPCR)

DRIP and DRIP-seq were performed as described (*Ginno et al., 2012*; *Sanz et al., 2016*). Briefly, after gentle genomic extraction and restriction enzyme fragmentation (HindIII, XbaI, EcoRI, SspI, BrsGI), 4 μg of digested DNA were incubated with 2 μg of S9.6 antibody overnight at 4°C in DRIP buffer (10 mM NaPO4 pH 7.0, 140 mM NaCl, 0.05% Triton X-100). In vitro RNase H digestion was used to generate a negative control. After incubation, antibody-DNA complexes were bound to Protein G Dynabeads and thoroughly washed, DNA was recovered with Chelex-100, the bound fraction was suspended in 0.1 ml 10% Chelex-100 (Bio-Rad), vortexed, boiled for 10 min, and cooled to room temperature (RT). This sample was added to 4 μl of 20 mg/ml Proteinase K followed by incubation at 55°C for 30 min while shaking. Beads were boiled for another 10 min. Sample was centrifuged and supernatant collected. Beads were suspended in 100 μl 2× TE, vortexed, centrifuged, and supernatants pooled. qPCR was performed with SYBR Select Master Mix (Thermo Fisher) and analyzed on a Light Cycler 480 (Roche), enrichment calculated by ratio of DRIP/input. For DRIP-seq analyses, libraries were prepared as described in *Ginno et al., 2012*; *Sanz et al., 2016*.

For dot blot analysis, fragmented genomic DNA was spotted in serial twofold dilutions and loaded onto a dot blot device assembled with Whatman paper and nitrocellulose membrane. The membrane was crosslinked in UV-crosslinker 120,000 uJ/cm² for 15 s, blocked with odyssey TBST blocking buffer, and incubated with the anti-DNA–RNA hybrid S9.6 antibody overnight at 4°C. Goat anti-mouse Alexa Fluor 680 secondary antibody (A21057) was used. Quantification on scanned image of blot was performed using ImageJ Lab software.

## Real-time quantitative RT-PCR

Total RNA was extracted from stimulated primary B cells with TRIzol (Invitrogen), followed by reverse transcription with ProtoScript II First Strand cDNA Synthesis Kit (NEB) according to the manufacturer's protocol. qPCR was performed using SYBR Select Master Mix (Thermo Fisher) and analyzed on a Light Cycler 480 (Roche). Gene of interest/ normalizing gene values ± SD were then normalized to the WT controls; germline switch region transcription values were normalized to CD79b transcripts before normalized to the WT control. Primers are listed in *Supplementary file 2*.

## Isolation and flow cytometry of bone marrow B cell progenitors

BM was isolated and resuspended in staining buffer as described in *Amend et al., 2016*. Erythrocytes were lysed, then cells were stained with anti-CD19 (APC; clone 1D3), anti-CD43 (PE-Cy7, clone S7), IgM (PE, clone R6–60.2), anti-CD11b (clone M1/70), and anti-B220 (FITC, clone RA3-6B2) (all from BD Biosciences) as described previously (*Mandal et al., 2019*). Counting beads (Spherotech, ACBP-50-10) were added to samples prior to analysis for cell quantitation. The absolute number of B cells at different stages of development was calculated by (number of events for the test samples/ number of events for the counting beads particle) * (number of beads particle used/volume of test sample initially used). Pre-pro-B cells are defined as B220+CD19- CD43+IgM-, pro-B cells are defined as B220+CD19+CD43+IgM-, large and small pre-B cells are defined as B220+CD19+CD43−IgM−FSChi and B220+CD19+CD43−IgM−FSClo, respectively, and immature B cells are defined as B220+CD19+CD43−IgM+ FSC, forward scatter. CD11B+ gating quantifies the myeloid compartment.

## Flow cytometry

Primary B cells were washed with PBS and stained with B220-FITC and biotin anti-IgG1 (BD), biotin anti-IgG2b (BioLegend), and PE anti-IgA (SouthernBiotech). For biotinylated primary antibodies, cells were then stained with PE-Streptavidin (Beckman Coulter). Data were collected on a BD FACSCanto and analyzed using FlowJo software. At least 20,000 events of live lymphoid cells were recorded. For CFSE staining, freshly isolated primary B cells were washed and resuspended in 0.1% BSA/PBS at $1 \times 10^7$ cells/ml and labeled with CFSE at a final concentration of 5 uM for 10 min at 37°C. CFSE was quenched with ice-cold RPMI 1640 medium containing 10% FCS and washed twice with BCM. Labeled cells were then cultured in BCM and appropriate stimuli for 72 or 96 hr days prior to analysis.

For cell cycle analysis, B cells were harvested 72 hr post-stimulation, washed once with PBS, and resuspended in ice-cold 70% ethanol while slowly vertexing. Cells were fixed overnight, then washed with PBS one time. Cell pellets were resuspended in propidium iodide (PI) staining solution (50 µg/ml PI and 100 units/ml RNase A in PBS), then incubated at RT in the dark for 2 hr. 50,000 gated events were collected on a BD FACSCanto and analyzed using FlowJo software.

## Metaphase chromosome preparation and FISH

The metaphase chromosome preparation and FISH were performed as described (*Waisertreiger et al., 2020*). Briefly, day 3 stimulated primary B cells were arrested in metaphase by a 1 hr treatment with 0.1 µg/ml demecolcine (Sigma, D1925), treated with 0.075 M KCl, fixed in methanol:acetic acid (3:1), spread onto glass slides, and air-dried. FISH was performed on metaphase cells using IgH probe. Prior to hybridization, slides were briefly heated over an open flame, denaturing DNA for IgH detection. Slides were washed in 1× PBS at RT for 5 min, post-fixed in 1% formaldehyde at RT for 5 min, and washed in 1× PBS at RT for 5 min. Slides were dehydrated in ethanol (75, 85, and 100%) at RT for 2 min each and air-dried. Cells and probes were co-denatured at 75°C for 3 min and incubated overnight at 37°C in a humid chamber. Slides were washed post-hybridization in 0.4 × SSC/0.3% NP-40 at 72°C (2 min), then 2 × SSC/0.1% NP-40 at RT (2 min). Slides were probed with 0.25 µM telomere probe (PNA Bio, F1002) for 1 hr at RT. Slides were then washed in 1× PBST (1× PBS, 0.5% Triton-X-100) for 5 min at 37°C. After wash with PBS and dehydrated in ethanol (75, 85, and 100%), slides were counterstained with Vectashield mounting medium containing DAPI (Vector Laboratories Inc, H-1200) before microscopy.

## Microscopy and analysis

B cells were isolated and cultured from a separate mouse for each experiment. A minimum of 50 metaphases were analyzed for each experiment. Metaphases images were acquired using an epifluorescent

Nikon microscope with NIS Elements AR4.40.00 software (Nikon). Downstream analysis used ImageJ software (NIH). *IgH aberration analysis*: DNA breaks were classified as 'at IgH' only if the BAC hybridized to the end of the break AND not co-localizing with the telomere probe (*Figure 3C*, second row from top). Alternatively, IgH BAC signal clearly fused to another chromosome was also considered rearrangement involving IgH (*Figure 3C*, third row from top). Rearrangements involving chromosome 12 (positive for BAC signal) but with fusions at other parts of the chromosome were considered 'non-IgH' rearrangements (*Figure 3C*, fourth row from top). Metaphase spreads with only one chromosome positive for IgH and another with a telomere break with no BAC signal likely had a break at IgH, but this cannot be confirmed without another probe for chromosome 12; therefore they were not counted as IgH breaks.

## Protein blot and immunoprecipitation

Protein expression was analyzed 72 hr post-stimulation for switching cells unless otherwise indicated. Briefly, 0.5 million cells were suspended with RIPA buffer (50 mM Tris–HCl pH 8.0, 150 mM NaCl, 2 mM EDTA pH8.0, 1% NP-40, 0.5% sodium deoxycholate, 0.1% SDS, Protease Inhibitors) and incubated at 4°C for 30 min, and following centrifuge to remove the thick DNA, the whole-cell lysis was boiled in protein loading buffer for SDS-PAGE. Immunoblotting was performed with the appropriate primary and secondary antibodies. For FLAG-RNaseH2B immunoprecipitation, total cell lysates were prepared in RIPA buffer and incubated with a desired antibody and appropriate protein A/G-agarose beads at 4°C overnight with gentle agitation. Beads were washed three times with lysis buffer, and immunocomplexes were eluted by boiling in SDS sample buffer for 5 min before loading. Anti-AID (Thermo Fisher ZA001) at 1:500 dilution, anti-β-actin (ABclonal, AC026) at 1:100,000 dilution, anti-FLAG (ABclonal, AE005), goat anti-mouse Alexa Fluor 680 secondary antibody (A21057), and goat anti-rabbit Alexa Fluor 790 secondary antibodies (A11367) were employed.

## ChIP

CHIP was performed as described (*Barlow et al., 2013*), Briefly, $1 \times 10^7$ stimulated primary B cells were cross-linked with 0.5% formaldehyde for 5 min, then quenched by addition of 125 mM glycine for 5 min at RT. Crosslinked cells were washed with ice-cold PBS three times and then resuspended in ice-cold RIPA buffer (50 mM Tris–HCl pH 8.0, 150 mM NaCl, 2 mM EDTA pH 8.0, 1% NP-40, 0.5% sodium deoxycholate, 0.1% SDS, Protease Inhibitors). Chromatin was sheared with a Bioruptor (Diagenode) ultrasonicator to the size range between 200 and 1000 bp. Samples were centrifuged and supernatant collected, 1% lysate as whole-cell DNA input. Antibody-coupled Dynabeads Protein G (Thermo Fisher) were used for immunoprecipitations performed overnight at 4°C. Anti-AID (Thermo Fisher ZA001), anti-Pol ll Serine5 Phospho (4H8, AB5408), anti-KU70/80 (Thermo MA1-21818), anti-RAD52 (ABclonal A3077), and IgG (AB37355) were used for immunoprecipitation. AID, KU70/80, and RAD52 ChIPs were performed on chromatin harvested 60 hr post-stimulation (*Robert et al., 2015*). Beads were washed once in each of the following buffers for 10 min at 4°C: low-salt buffer (0.1% SDS, 1% Triton X-100, 2 mM EDTA, 20 mM Tris–HCl pH 8.0, 150 mM NaCl), high-salt buffer (0.1% SDS, 1% Triton X-100, 2 mM EDTA, 20 mM Tris–HCl pH 8.0, 500 mM NaCl), LiCl buffer (0.25 M LiCl 1% NP-40, 1% sodium deoxycholate, 1 mM EDTA, 10 mM Tris–HCl pH 8.0), and TE buffer (50 mM Tris pH 8.0, 10 mM EDTA). DNA was recovered with Chelex-100 and analysis by qPCR. Data were analyzed using the comparative CT method. Fold enrichment was calculated as ChIP/input. Primers used for qPCR are listed in *Supplementary file 2*.

## Alkaline gel electrophoresis

For alkaline gel electrophoresis, genomic DNA was extracted from day 3 LPS/IL-4/α-RP105-stimulated B cells. Then, 2 ug of genomic DNA was incubated in 0.3 M NaOH for 2 hr at 55°C and separated on an 0.9% agarose gel (50 mM NaOH, 1 mM EDTA) as previously described (*McDonell et al., 1977*). Gels were neutralized with neutralizing solution (1 M Tris–HCl, 1.5 M NaCl) and stained with ethidium bromide prior to imaging. Densitometry was analyzed using ImageJ software (NIH). Genomic DNA samples were also analyzed by native gels (0.9% agarose gel in 1× TAE) to quantify DNA fragmentation in the absence of alkaline activity.

## Retroviral preparation and B cell infection

Viral infection was performed as described (*Waisertreiger et al., 2020*). After verification of infection efficiency by flow cytometry, cells were harvested for genomic DNA extraction, DRIP, immunoprecipitation-WB, and FISH.

## Junction analysis

Sμ-Sγ1 switch junctions were amplified using published primers (*Zan et al., 2017*). Briefly, genomic DNA was prepared from 72 hr LPS/IL-4/α-RP105-stimulated B cells. PCR products were cloned using pGEM-TA cloning kit (Promega) and sequenced with T7/SP6 universal primers. Sequence analysis was performed using the Snap gene software. Junction sequences were compared against 129Sv and C57BL/6 backgrounds, and the comparison showing the fewest alterations was chosen for final analyses.

## LAM-HTGTS library preparation and analysis

Genomic DNA was isolated from WT, *Setx⁻/⁻*, *Rnaseh2b^f/f^*, and *Setx⁻/⁻Rnaseh2b^f/f^* cells 72 hr after stimulation to switch to IgG1 with 72 hr LPS/IL-4/α-RP105. LAM-HTGTS libraries were prepared from genomic DNA as previously described (*Hu et al., 2016*; *Yin et al., 2019a*; *Yin et al., 2019b*). Briefly, isolated genome DNA was sonicated with a bioruptor to generate 0.2–2 kb size fragments with a peak at approximately 750 bp. Sonicated DNA was annealed and extended with a biotin primer to Sμ (5′/5BiosG/CAGACCTGGGAATGTATGGT3′). The DNA was denatured at 95°C 5 min and cooled down on ice for 5 min, then the biotinylated PCR product was purified with Dynabeads MyOne T1 (Thermo Fisher) and used to perform an on-bead ligation of the adaptor. An Sμ-specific nested primer (5′CACACAAAGACTCTGGACCTC3′) and the adaptor-specific primer were used to do the nested PCR, and AflII was used to remove germline sequence product. After amplification with P5 P7 adaptor primer, PCR products were run on 1% TAE gel and products 500–1000 bp in size were cut out and isolated by gel extraction. The purified DNA underwent Ampure bead double-size selection before sending for QC and Mi-seq sequencing. Sequences were aligned to custom genomes, substituting the mm9 sequence from (114, 494, 415-114, 666, 816) with sequence from the NG_005838.1 (GenBank accession no. NG_005838.1) C57BL/6 IgH sequence on chr 12 (to 11,172–183,818), or sequence from the AJ851868.3 (GenBank accession no. AJ851868.3) S129 IgH sequence (1,415,966–1,592,715). Sequences were analyzed as detailed in *Crowe et al., 2018*; *Hu et al., 2016* with the following adjustments. Junctions were analyzed by comparing sequences to both C57BL/6 and S129 backgrounds. A consensus genome was generated between C57BL/6 and S129 that identified switch variants for both backgrounds. If reads fell into either category, 100% identity with either C57BL/6 or S129, they were deemed not mutated—this eliminated overestimation of insertions, deletions, and mutations. Here, we use 'mutation' to define the alteration of one or more nucleotides from either original sequence without changing the overall spacing. Of note, no junctions aligned to S129 on one side and C57BL/6 on the other, indicating all switching occurred between a single allele. For all reads, 50 bp of sequence upstream and downstream the junction were analyzed. Junction sequences referred to as 'blunt' have perfect match with bait (Sμ region) and prey (Sγ1 region) on both flanking sides, and a blunt join (no potential MH at junction site). Deletions are defined as regions missing nucleotides adjacent to prey-break site but having 100% homology in flanking regions. Insertions are defined as regions containing nucleotides that map to neither the bait nor the prey-break site. MHs are defined as regions of 100% homology between the bait and the prey-break site. Blunt junctions are considered to have no MHs or insertions. Junctions could include more than one event class (insertion, deletion, mutation, MH) having a more complex result; a junction exhibiting MH may also have a deletion or mismatches in flanking DNA. Thus, junction types were quantified on whether they contained at least one event of the class listed in the header (deletions, insertions, MHs). Additional scripts used in analysis can be found in GitHub (https://github.com/srhartono/TCseqplus, *Hartono, 2022*; copy archived at swh:1:rev:d3b042f06d24e2fe18144db42029ff79a922d0b8).

## Acknowledgements

This work was supported by research funding from the National Cancer Institute K22CA188106, University of California Cancer Research Coordinating Committee (UC-CRCC) seed grant CRR-20-635379,

and National Institute for General Medical Studies grants R01 GM134537 (JHB); R21 AI151610 (RS); and R35 GM139549 (FC). The sequencing was performed by the DNA Technologies and Expression Analysis Cores at the University of California Davis Genome Center, supported by National Institutes of Health Shared Instrumentation Grant (S10 OD010786-01). This study utilized the University of California Davis Cancer Center Flow Cytometry core partially supported by National Institute of Health grant S100D018223. Thanks to Drs. Klaus Rajewsky, Martin Lavin, and Axel Roers for mouse models. We would like to thank Dr. Commodore St Germain and all members of the Barlow and Chedin labs for helpful discussions and suggestions, and Jack McTiernan for assistance with figure design.

## Additional information

### Funding

| Funder | Grant reference number | Author |
|---|---|---|
| National Cancer Institute | CA188106 | Jacqueline Barlow |
| National Institute of General Medical Sciences | GM134537 | Jacqueline Barlow |
| National Institute of General Medical Sciences | GM139549 | Frédéric Chédin |
| National Institute of Allergy and Infectious Diseases | AI151610 | Roger Sciammas |
| University of California Cancer Research Coordinating Committee (UC- CRCC) | Seed Grant CRR-20-635379 | Jacqueline Barlow |

The funders had no role in study design, data collection and interpretation, or the decision to submit the work for publication.

### Author contributions

Hongchang Zhao, Formal analysis, Validation, Investigation, Visualization, Writing – original draft, Writing – review and editing; Stella R Hartono, Data curation, Software, Validation, Visualization; Kirtney Mae Flores de Vera, Krishni Satchi, Tracy Zhao, Lionel Sanz, Investigation; Zheyuan Yu, Data curation, Formal analysis, Validation; Roger Sciammas, Formal analysis, Supervision, Methodology, Writing – review and editing; Frédéric Chédin, Resources, Supervision, Writing – review and editing; Jacqueline Barlow, Conceptualization, Supervision, Funding acquisition, Visualization, Writing – original draft, Project administration, Writing – review and editing

### Author ORCIDs
Jacqueline Barlow http://orcid.org/0000-0002-9042-6245

### Ethics

This study was performed in strict accordance with the recommendations in the Guide for the Care and Use of Laboratory Animals of the National Institutes of Health. All of the animals were handled according to approved institutional animal care and use committee (IACUC) protocols (#21828) of the University of California Davis.

### Decision letter and Author response
Decision letter https://doi.org/10.7554/eLife.78917.sa1
Author response https://doi.org/10.7554/eLife.78917.sa2

## Additional files

### Supplementary files
• Supplementary file 1. Summary of individual fluorescent in situ hybridization (FISH) results for experiments shown in *Figures 3 and 5*.

- Supplementary file 2. Primers used in qPCR and junction analysis. Source data files
- MDAR checklist

## Data availability

HTGTS-Seq data has been deposited to the Gene Expression Omnibus (GEO) database (GEO accession GSE201210). All data was made publicly available upon acceptance.

The following dataset was generated:

| Author(s) | Year | Dataset title | Dataset URL | Database and Identifier |
|---|---|---|---|---|
| Zhao H, Hartono S, De Vera K, Yu Z, Satchi K, Zhao T, Sciammas R, Sanz L, Chedin F, Barlow J | 2022 | Senataxin and RNase H2 act redundantly to suppress genome instability during class switch recombination | https://www.ncbi.nlm.nih.gov/geo/query/acc.cgi?acc=GSE201210 | NCBI Gene Expression Omnibus, GSE201210 |

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
