## [Editor Report]

R loops have been described at the immunoglobulin heavy chain (Igh) locus long ago. However, their contribution to Igh diversification by class switch recombination (CSR) and locus integrity has been elusive. The authors show that R loop removal by the activity of senataxin and RNase H2 does not influence CSR but is required to suppress genome instability at the Igh locus. This article will be of interest to the audience in the fields of genome integrity and B lymphocyte biology.

---

## [Decision Letter]

**Decision letter after peer review:**

Thank you for submitting your article "Senataxin and RNase H2 act redundantly to suppress genome instability during class switch recombination" for consideration by *eLife*. Your article has been reviewed by 3 peer reviewers, including Michela Di Virgilio as Reviewing Editor and Reviewer #1, and the evaluation has been overseen by Betty Diamond as the Senior Editor. The following individual involved in review of your submission has agreed to reveal their identity: Rushad Pavri (Reviewer #2).

Essential revisions:

1) Cd19-Cre acts during B cell development in the bone marrow and hence it is important to know whether the phenotypes observed in mature B cells were influenced by the early KO of Setx and Rnaseh2. Is B cell development affected in the mutant mice?

2) In line with the point above, what are the level of R loops before (resting B cells) and after B cell activation in the single- and double-mutants compared to WT? The experiments performed in Figure 4 do not address this question but rather proves that the increased R loop formation is indeed caused by the enzyme loss.

3) A major concern is that the variability of several key experiments (e.g. Figure 1D, 4D, 4E, 5-SFigure 1, D and E) is really high, and should be minimized to be able to draw correct conclusions. Furthermore, in Figure 1E, the levels of germline transcription are assessed at 72 h post-activation, at which point the Igh locus has already undergone major structural changes in a big portion of the cell population. Analysis should be provided at 48 h post-activation max.

4) The authors should discuss why they feel the combined deficiency of senataxin and RNase2b in activated B cells only causes an increase in R loops in Sm, but not in Sg1.

5) Deletion of both R-loop resolving enzymes does not lead to any observable effect in CSR to all tested isotypes (Figure 2). However, the use of RP105 could potentially mask time-sensitive phenotypes since the proliferation rate is highly increased. What is the CSR phenotype of the single- and double-KO cells without RP105 in the cytokine cocktails?

6) Also, the authors do not address whether increased R loops skew the mutation spectrum towards the non-template strand, which would be expected if R loop removal was delayed. Is this the case?

7) According to the current model, lack of R loop removal leads to repair of the AID-induced breaks through A-EJ. In this regard, how can the authors justify the link between persistent R loop formation and an increase in A-EJ, when persistent R loop formation is only observed in the double-KO cells, but altered junction profiles are observed in both Setx-/- cells and double-KO (Figure 6 and Figure 6-SFigure 1 C)?

8) Although quite an interesting observation, there is no data in the manuscript that hints at A-EJ apart from analysis of junction. Does knock-out/knock-down of A-EJ factors such as pol theta affect insertional repair in the KO? Is the recruitment of known A-EJ factors increased? In the same line and as proposed by the authors themselves, is the recruitment of cNHEJ factors reduced?

[Editors’ note: further revisions were suggested prior to acceptance, as described below.]

Thank you for resubmitting your work entitled "Senataxin and RNase H2 act redundantly to suppress genome instability during class switch recombination" for further consideration by *eLife*. Your revised article has been evaluated by Betty Diamond (Senior Editor) and a Reviewing Editor.

The manuscript has been improved but there are some remaining issues that need to be addressed, as outlined below:

Major comment #3: It is possible that high variability exists between biological replicates of RNaseH2B ChIP, especially conducting these experiments in primary cells. They could show the results of the 4 individual experiments in separate plots in the supplemental figure.

Major comment #8: The authors nicely showed in new experiments by ChIP that Ku70/Ku80 binding at Sμ is reduced in double KO cells (~50% of WT). This is consistent with increased R-loops at Sμ interfering with KU binding but does not seem to track with CSR defect observed in KU-deficient cells. The authors could discuss why reduced KU binding in the double KO cells may not lead to a CSR defect.

For the additional text added or changed in the revised manuscript, please indicate the line numbers of the changes to each comment.

Some formatting errors:

Figure 3 figure supplement 1B: check the Y-axis labeling (eg. B220 should be B220+, CD43 should be CD43+…); 1D: X-axis (myeloid).

Figure 6A, label "% of Sμ-Sγ1 junction reads with at least 1 ": the "l" of "least" was deleted in the revised figure version. Similar letter/symbol omissions are also present in the labels of panels A, B and D of Figure 6—figure supplement 1.

*Reviewer #1 (Recommendations for the authors):*

The authors have provided several new experiments and added more extended explanations in the revised manuscript, which have collectively addressed the majority of points raised by this (and other) reviewers and substantially improved the study.

*Reviewer #2 (Recommendations for the authors):*

I am satisfied with the revisions. I do not have any outstanding major questions or concerns, although I would have liked to see the mutation analysis (via Sanger to deep sequencing) to see if mutation asymmetry at Smu was observed in the dKO cells.

*Reviewer #3 (Recommendations for the authors):*

The authors have addressed our original comments and have improved the manuscript, which is now recommended for publication. They may wish to address the following points prior to publication.

Major comment #3: It is possible that high variability exists between biological replicates of RNaseH2B ChIP, especially conducting these experiments in primary cells. They could show the results of the 4 individual experiments in separate plots in the supplemental figure.

Major comment #8: The authors nicely showed in new experiments by ChIP that Ku70/Ku80 binding at Sμ is reduced in double KO cells (~50% of WT). This is consistent with increased R-loops at Sμ interfering with KU binding but does not seem to track with CSR defect observed in KU-deficient cells. The authors could discuss why reduced KU binding in the double KO cells may not lead to a CSR defect.

---

## [Author Response]

Essential revisions:1) Cd19-Cre acts during B cell development in the bone marrow and hence it is important to know whether the phenotypes observed in mature B cells were influenced by the early KO of Setx and Rnaseh2. Is B cell development affected in the mutant mice?

We have added bone marrow analyses that include quantitation of pre- and pro-B cell numbers in the bone marrow – new (figure 3—figure supplement 1). We were assisted in experimental design, execution, and analysis by Dr. Roger Sciammas, an expert in B cell development. We analyzed pre-pro-B, pro-B, small and large pre-B, immature B and myeloid progenitors by flow cytometry using anti-CD19 (APC; clone 1D3), antiCD43 (PE-Cy7, clone S7), IgM (PE, clone R6–60.2), Anti-CD11b; (clone M1/70) and anti-B220 (FITC, clone RA36B2) (all from BD Biosciences) and counting beads (Spherotech, ACBP-50-10) to assess absolute numbers. All bone marrow cell progenitors examined showed similar numbers between all four genotypes, indicating that germline loss of *Setx* and/or conditional deletion of *Rnaseh2b* do not affect B cell development. Mature naïve splenic B cells were also similar between the 4 genotypes.

2) In line with the point above, what are the level of R loops before (resting B cells) and after B cell activation in the single- and double-mutants compared to WT? The experiments performed in Figure 4 do not address this question but rather proves that the increased R loop formation is indeed caused by the enzyme loss.

We performed DRIP in resting B lymphocytes and obtained results largely similar to 72 hours, therefore the difference in R loop formation at Sμ is already observable in G0. The new DRIP data from resting cells is presented in Figure 1 —figure supplement 1C.

3) A major concern is that the variability of several key experiments (e.g. Figure 1D, 4D, 4E, 5-SFigure 1, D and E) is really high, and should be minimized to be able to draw correct conclusions. Furthermore, in Figure 1E, the levels of germline transcription are assessed at 72 h post-activation, at which point the Igh locus has already undergone major structural changes in a big portion of the cell population. Analysis should be provided at 48 h post-activation max.

We performed RT-qPCR in cells 48 post-stimulation and obtained results largely similar to 72 hours. Imu-Cmu is reduced in DKO similar to 72h; at 48h Ig1-Cg1 is also significantly down in DKO by about 50%. The trend was observable at 72 h but was not significant. The new 48h RT-qPCR data is presented in Figure 1E, while 72 h results have been moved to Figure 1 figure supplement 1D.

The AID ChIP in figure 5 supplement 1D was performed on chromatin harvested at 60 hours post-stimulation similar to other published studies. This time point after peak AID expression but ~12h prior to high levels of successful CSR product formation can be observed. Apologies the timing of AID ChIP was not clear in the original manuscript; we have now added this information to the text, figure legend and methods.

While the RNase H2 ChIP indeed shows variation (Figure 4D and E), this is at least partially due to the fact that retroviral reconstitution in primary cells is inherently variable. It is not possible to alter the timing of this experiment due to the constraints of protein expression by retroviral reconstitution as cells must be proliferating to be infected (two rounds of spin-fection at 24 and 48h post-stimulation), then require a minimum of 24h for protein expression. We harvest chromatin (for ChIP) or genomic DNA (for DRIP) at 96 h as the infection process delays CSR and allows for the introduced protein to function for longer.

Unfortunately the variability we observe in ChIP experiments is likely due to the fact that we are analyzing dynamic processes at small time windows; and the natural variation from mouse to mouse, stimulation to stimulation, and infection to infection. We cannot discard the results presented as the experiments were performed in mice were matched as closely as possible to be similar age and sex using littermates when possible. Due to the complexity of the genetics, this was not always feasible and analysis of experiments using littermates vs. mice from other litters showed no difference in variability. Experiments shown had no clear failures (control experiment failures, poor switching, genotyping issues, etc) that would warrant discarding the results.

4) The authors should discuss why they feel the combined deficiency of senataxin and RNase2b in activated B cells only causes an increase in R loops in Sm, but not in Sg1.

We have added text to the discussion on possible reasons why we observe this discrepancy.

5) Deletion of both R-loop resolving enzymes does not lead to any observable effect in CSR to all tested isotypes (Figure 2). However, the use of RP105 could potentially mask time-sensitive phenotypes since the proliferation rate is highly increased. What is the CSR phenotype of the single- and double-KO cells without RP105 in the cytokine cocktails?

To investigate if α-RP105 potentially obscures subtle differences in CSR frequency between WT and cells lacking Setx or RNase H2, we stimulated WT, *Setx^-/-^*, *Rnaseh2b^f/f^*, and *Setx^-/-^Rnaseh2b^f/f^* cells with LPS+IL-4 alone and assessed CSR by flow cytometry at 72 hours. We have performed 5X, and found no statistical difference between the four genotypes, indicating the decrease in cell death / alteration in proliferation by addition of α-RP105 does not mask any differences in CSR between these genotypes. The new results for 5 independent runs are summarized in Figure 2 figure supplement 1.

6) Also, the authors do not address whether increased R loops skew the mutation spectrum towards the non-template strand, which would be expected if R loop removal was delayed. Is this the case?

This is an exciting and interesting question, but unfortunately cannot be answered using LAM-HTGTS data as libraries specifically amplify a single strand of the genomic DNA by the use of a biotinylated primer for purification (Frock et al. 2016). Briefly, the biotinylated primer anneals to the bait sequence up to 400 bp upstream of the break site and extension across the template (potentially containing a junction event). PCR extension products specific to the bait sequence are denatured and then purified by streptavidin. the DNA without the streptavidin tag is lost. The retained biotin-ssDNA molecule is ligated with the adaptor and then amplified using a primer to the ligated adapter sequence and nested PCR primers which are located downstream of biotin-primer. This newly synthesized DNA is purified away from the streptdavidin bead. Thus all sequencing results originate from a single strand of DNA—the strand extended from the biotin primer.

7) According to the current model, lack of R loop removal leads to repair of the AID-induced breaks through A-EJ. In this regard, how can the authors justify the link between persistent R loop formation and an increase in A-EJ, when persistent R loop formation is only observed in the double-KO cells, but altered junction profiles are observed in both Setx-/- cells and double-KO (Figure 6 and Figure 6-SFigure 1 C)?

Indeed, cells lacking Setx alone also exhibit changes in sequence junctions as the referees note. However these results often to not reach statistical significance, unlike double-deficient cells. The predominant event observed in single knockout cells is an increase in insertion events by Sanger sequencing in Figure 6C. Here we presume this is due to the fact that many fewer molecules were sequenced, and 25% of 36 events is 9 sequneces. Regardless, we agree that loss of Setx alone likely has an effect but unfortunately it remains below our level of statistical significance in DRIP, ChIP and FISH experiments and additional repeats are not likely to change this outcome. We have added these points to the Discussion section to highlight this possibility.

8) Although quite an interesting observation, there is no data in the manuscript that hints at A-EJ apart from analysis of junction. Does knock-out/knock-down of A-EJ factors such as pol theta affect insertional repair in the KO? Is the recruitment of known A-EJ factors increased? In the same line and as proposed by the authors themselves, is the recruitment of cNHEJ factors reduced?

We agree this point is interesting and have repeatedly attempted to develop it using a variety of approaches. We performed ChIP with a commercially available antibody for Ku70/80 and found a reduction in Ku70/80 association at Sμ but not Sg1 in DKO cells, consistent with DRIP results (new Figure 6D ). Ku70/80 association was also somewhat reduced in the single mutants, but it was not statistically significant. Ku70/80 competes with the homologous recombination protein Rad52 at switch regions during CSR, and these proteins potentially recognize different substrates. To assess if Rad52 binding is increased, we also measured Rad52 recruitment to switch regions by ChIP. We saw an increase in Rad52 binding, however the increase was not statistically significant (new Figure 6E). We have added text to the results and discussion for these new experiments.

Negative results for referees: We attempted to inhibit Pol theta with a published inhibitor but could not validate Pol theta/alternative NHEJ as the target of the inhibitor. Further, shRNA-mediated knock down of Pol theta in primary cells by retroviral infection is unlikely to yield a measurable effect due to experimental constraints of primary B cells. Finally, combining genetic knockout of Pol theta in the *Setx^-/-^ RNaseh2b Cd19cre* background is prohibitively expensive.

We have also performed multiple additional ChIPs to DNA repair proteins to measure relative recruitment with variable results. Commercially available CtIP (2) and Msh2 (1) antibodies as well as two different Ku70 antibodies have not been successful in our IP or ChIP experiments. There are no commercially available antibodies against pol theta rated for ChIP or IP.

[Editors’ note: further revisions were suggested prior to acceptance, as described below.]

The manuscript has been improved but there are some remaining issues that need to be addressed, as outlined below:Major comment #3: It is possible that high variability exists between biological replicates of RNaseH2B ChIP, especially conducting these experiments in primary cells. They could show the results of the 4 individual experiments in separate plots in the supplemental figure.

We have added a new figure -- Figure 4 supplemental figure 1 -- which contains the 4 independent repeats of the FLAG-RNase H2B ChIP experiment.

Line 894-895: We have added a figure legend description for the new supplemental figure.

Major comment #8: The authors nicely showed in new experiments by ChIP that Ku70/Ku80 binding at Sμ is reduced in double KO cells (~50% of WT). This is consistent with increased R-loops at Sμ interfering with KU binding but does not seem to track with CSR defect observed in KU-deficient cells. The authors could discuss why reduced KU binding in the double KO cells may not lead to a CSR defect.

A reduction in Ku binding specifically at Smu is not likely to dramatically impact CSR as DSBs are not limiting. Further, alternative repair pathways may be able to salvage some EJ events.

We have added two sentences describing this in the results: line 397-398:

“Similarly, the reduction in Ku70/80 binding specifically at Sμ is also unlikely to have a major impact on CSR frequency (Figure 6D).”

line 412-414:

“Indeed, repair by alternate pathways may contribute to why we observe not major defect in CSR though Ku70/80 binding is reduced in *Setx^-/-^Rnaseh2b^f/f^* cells.”

Some formatting errors:Figure 3 figure supplement 1B: check the Y-axis labeling (eg. B220 should be B220+, CD43 should be CD43+…); 1D: X-axis (myeloid)

We apologize for this issue. The superscript + marks were deleted in PDF building possibly due to their small size. The same is true for the misspellings in the immature (“immature”) and myeloid (“Mye o d”) labeling where lowercase Ls and Is did not appear. To limit this issue, we have increased font sizes and re-converted illustrator files to PDFs at the highest resolution.

Figure 6A, label "% of Sμ-Sγ1 junction reads with at least 1 ": the "l" of "least" was deleted in the revised figure version. Similar letter/symbol omissions are also present in the labels of panels A, B and D of Figure 6—figure supplement 1.

We apologize for this issue, it was a problem with PDF conversion (see above).